# MDP: A Generalized Framework for Text-Guided Image Editing by Manipulating the Diffusion Path

**Qian Wang**                                            *qian.wang@kaust.edu.sa*
*King Abdullah University of Science and Technology*

**Biao Zhang**                                           *biao.zhang@kaust.edu.sa*
*King Abdullah University of Science and Technology*

**Michael Birsak**                                       *michael.birsak@kaust.edu.sa*
*King Abdullah University of Science and Technology*

**Peter Wonka**                                          *peter.wonka@kaust.edu.sa*
*King Abdullah University of Science and Technology*

**Reviewed on OpenReview:** *https://openreview.net/forum?id=2666*

## Abstract

Image generation using diffusion can be controlled in multiple ways. In this paper, we systematically analyze the equations of modern generative diffusion networks to propose a framework, called **MDP**, that explains the design space of suitable manipulations. We identify 5 different manipulations, including intermediate latent, conditional embedding, cross attention maps, guidance, and predicted noise. We analyze the corresponding parameters of these manipulations and the manipulation schedule. We show that some previous editing methods fit nicely into our framework. Particularly, we identified one specific configuration as a new type of control by manipulating the predicted noise, which can perform higher-quality edits than previous work for a variety of local and global edits.

## 1 Introduction

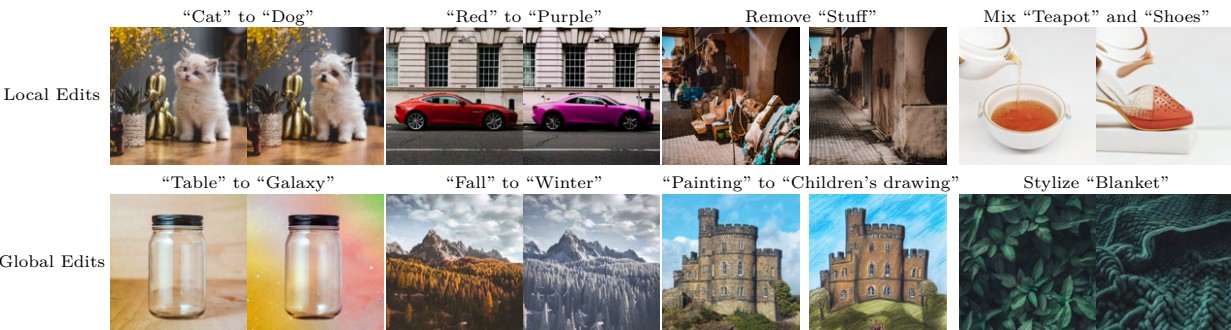

Figure 1: Our proposed manipulation of predicted noise, which fits into our proposed editing framework **MDP**, can do multiple local and global edits without training or finetuning.

Previously, image editing using GANs has achieved great success (Isola et al., 2017; Zhu et al., 2017; Choi et al., 2018; Wang et al., 2018; Huang et al., 2018; Kim et al., 2017; Bau et al., 2020). As large-scale text-to-image datasets became available, text-guided image synthesis and editing has obtained increasing attention (Ramesh et al., 2021; Crowson et al., 2022; Ding et al., 2021; 2022; Gafni et al., 2022). Generative diffusion models Ho et al. (2020); Rombach et al. (2022); Saharia et al. (2022); Nichol et al. (2022); Ramesh

et al. (2022) are also a powerful tool for multiple image processing tasks, such as inpainting Lugmayr et al. (2022); Choi et al. (2021); Li et al. (2022); Xie et al. (2022), style transfer Kwon & Ye (2022); Bansal et al. (2023), text-guided image editing Brooks et al. (2022); Kawar et al. (2022); Hertz et al. (2022); Tumanyan et al. (2022), map-to-image translation Voynov et al. (2022); Avrahami et al. (2022a) and segmentation Burgert et al. (2022); Baranchuk et al. (2021).

We are interested in simple and effective image editing methods that do not require retraining, fine-tuning, or training of an auxiliary network. Specifically, for text-guided image editing, we can leverage pre-trained diffusion models to perform editing tasks on real images. An arbitrary real image can first be embedded into the diffusion latent space to obtain an initial noise tensor (Wallace et al., 2022; Mokady et al., 2022). Then, by manipulating the diffusion generation path starting from this initial noise tensor, we can edit the input image.

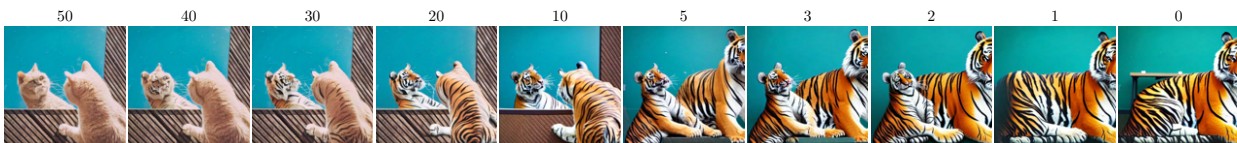

Figure 2: Assume the total number of sampling steps $T = 50$. The number on top of each image shows the number of diffusion generation steps for which we use $\mathbf{c}^{(A)}$ "*Photo of a cat sitting next to a mirror*" to invert and then denoise. For the remaining diffusion generation steps, we switch to $\mathbf{c}^{(B)}$ "*Photo of a tiger sitting next to a mirror*" to denoise. If we change condition in the early stage (e.g., 30 and 20), we can preserve the layout of $\mathbf{c}^{(A)}$ while including the semantics from $\mathbf{c}^{(B)}$. If we switch at a later stage (e.g., 5, 3, 2, 1), the layout of $\mathbf{c}^{(A)}$ is hardly maintained in the result.

Given an input image, we perform edits guided by a condition (for example a text prompt or a class label) while keeping the overall layout from the input image. We can see this editing task as combining the layout from the input image with new semantics from the condition. Since diffusion models generate images progressively, the layout of an image is generated during the early timesteps of the denoising process, while the semantics, i.e. the texture, color, details, are generated during later steps (Hoogeboom et al., 2023; Liew et al., 2022). Therefore, it is intuitive that each timestep should be controlled separately.

For each denoising step, there are several components that we can modify during the diffusion process. Intuitively, we want to denoise using information from the input image, for which we want to preserve the layout, in the early stages, and then the components of the new condition in the later stages. At this point, two questions arise: (1) *Which component of the generation process should be modified?* (2) *For which denoising steps should the modification take place?*

In order to tackle the problems, we analyze the equations of modern generative diffusion networks to find out which variables could be manipulated and identified 5 different manipulations that are suitable: intermediate latent, conditional embedding, cross attention maps, guidance and predicted noise. We take the manipulation of attention maps from the seminal paper Prompt-to-Prompt (P2P) Hertz et al. (2022). As result of the analysis, we introduce a generalized editing framework, **MDP** (**M**anipulating the **D**iffusion **P**ath), that contains these 5 different manipulations, their corresponding parameters, and a manipulation schedule for them. Some of the manipulations, though simple and have not been explored before, are proved to be meaningful for different editing applications.

We categorize the image editing applications into local editing and global editing. We find that different manipulations differ from the degree of preserving the original layout. If too much original layout is preserved, new semantics has difficulty injecting into the edited image; while if too less original layout is preserved, original layout has to be injected for more diffusion sampling steps, while original semantics inevitably leaks into the edited image in the later diffusion sampling steps.

In this design space, we identified one particular configuration, MDP-$\epsilon_t$, that yields very good results for a variety of local and global image editing problems shown in Figure 1. The results from MDP-$\epsilon_t$ are more consistent in quality than previous work (P2P) and other possible configurations. That means, MDP-$\epsilon_t$ has

the best degree of layout preservation compared to other manipulations. We advocate for the use of MDP-$\epsilon_t$ as a new method for diffusion-based image editing.

Our work makes the following contributions:

- We present a framework for a generalized design space to manipulate the diffusion path that includes multiple existing methods.

- We analyze the design space to find good configurations suitable for editing applications.

- We propose a new solution by manipulating predicted noise that can solve practical local and global editing problems better than other configurations and previous work.

## 2 Related Work

**Image diffusion models.** In recent years, the interest in deep generative models has gained strong momentum and many different methods have been proposed to create high-quality samples from a magnitude of different data domains. One kind of architecture that currently stands out are diffusion models (Ho et al., 2020; Dhariwal & Nichol, 2021; Karras et al., 2022).

In order to obtain better control over the denoising process and the generated content, several works proposed text-conditioned image synthesis using diffusion models and either CLIP guidance or classifier-free guidance (Nichol et al., 2022; Ramesh et al., 2022; Saharia et al., 2022).

Several approaches focus on different optimization strategies and either propose to generalize DDPMs via a class of non-Markovian diffusion processes to trade off computation for sample quality (Song et al., 2021) or to do the diffusion process in the latent space of a pretrained autoencoder (Rombach et al., 2022) instead of the RGB pixel space.

**Image editing with diffusion models.** InstructPix2Pix Brooks et al. (2022) can edit images by following user instructions. Paint by Example Yang et al. (2022) allows users to replace an object with a conditional example image. ControlNet Zhang & Agrawala (2023) trains a task-specific condition network on top of a pre-trained diffusion model and supports various edits based on a specific training set. All these works usually require re-designing of the model, collecting training data and a long training time to do image translation.

Blended DIffusion Avrahami et al. (2022b) and Blended Latent Diffusion Avrahami et al. (2023) adopt a user mask as input and design specific loss function to obtain good text-guided image editing results within the mask region. UniTune Kawar et al. (2022) and Imagic Kawar et al. (2022) can do realistic text-guided image editing by either just finetuning the diffusion models or the models and text embeddings together. However, because of the optimization process, it takes several minutes or more to generate a single image.

Recently, there are many interesting works that only utilize a pretrained diffusion model to do text-guided image editing without any training or finetuning (Meng et al., 2021; Radford et al., 2021; Couairon et al., 2022; Liew et al., 2022; Hertz et al., 2022; Park et al., 2022; Parmar et al., 2023; Tumanyan et al., 2022). DiffEdit Couairon et al. (2022) automatically generates a mask by computing the differences between the noisy latent generated from an input image and a conditional text prompt. However, because of its inpainting nature, DiffEdit cannot perform global editing operations like converting a photo to an oil painting. MagicMix Liew et al. (2022) interpolates the noisy input latent and the denoised latent to mix the objects that have large semantic differences. Prompt-to-Prompt Hertz et al. (2022) proposes to manipulate the cross-attention maps corresponding to the changes between the input and guided text prompt. Concurrent work pix2pix-zero Parmar et al. (2023) computes an editing direction to edit an image based on a provided prompt, combined with a cross-attention mechanism to preserve the layout of the input image.

Compared to prior works, we propose a generalization that includes multiple previous approaches as special case. We also highlight a novel manipulation, that is suitable to better perform a wide range of local and global edits than previous methods. Our edits do not require masks for an input image. Our method is purely based on a pre-trained diffusion model and does not require finetuning.

# 3 Design Space for Manipulating the Diffusion Path

We provide the details of sampling using diffusion model, image editing using diffusion model and inversion for diffusion model in the Supplementary Materials. Here we introduce the design space of our framework. We abstract our framework to linear interpolation operations and no higher-order operations are explored in this work.

Table 1: We use the superscript $^{(A)}$ to represent outputs obtained from condition $\mathbf{c}^{(A)}$ and $^{(B)}$ for condition $\mathbf{c}^{(B)}$. The edited outputs are represented with superscript $^{(\star)}$. $\omega$ is a number between 0 and 1. $\beta$ is usually a positive real number.

| | Pre | Changes | Post | Exist. Methods |
|---|---|---|---|---|
| Inter. Denoised Interp. Inter. Denoised Masking | $\left\{\mathbf{x}_t^{(A)} = \text{GenPath}\left(\mathbf{x}_T, \mathbf{c}^{(A)}, t\right)\right.$ | $\mathbf{x}_t^{(\star)} = \omega\mathbf{x}_t^{(A)} + (1-\omega)\mathbf{x}_t^{(\star)}$ $\mathbf{x}_t^{(\star)} = \mathbf{M}\odot\mathbf{x}_t^{(A)} + (1-\mathbf{M})\odot\mathbf{x}_t^{(\star)}$ | $\left\{\begin{array}{l}\boldsymbol{\epsilon}_t^{(\star)} = \boldsymbol{\epsilon}_\theta\left(\mathbf{x}_t^{(\star)}, \mathbf{c}^{(B)}, t\right) \\ \mathbf{x}_{t-1}^{(\star)} = \text{DDIM}\left(\mathbf{x}_t^{(\star)}, \boldsymbol{\epsilon}_t^{(\star)}, t\right)\end{array}\right.$ | Liew et al. (2022) Couairon et al. (2022) |
| Condition Emb. Interp. | $\left\{\mathbf{x}_t^{(A)} = \text{Gen}(\mathbf{x}_T, \mathbf{c}^{(A)}, t)\right.$ | $\mathbf{c}^{(\star)} = \omega\mathbf{c}^{(A)} + (1-\omega)\mathbf{c}^{(B)}$ | $\left\{\begin{array}{l}\boldsymbol{\epsilon}_t^{(\star)} = \boldsymbol{\epsilon}_\theta\left(\mathbf{x}_t^{(A)}, \mathbf{c}^{(\star)}, t\right) \\ \mathbf{x}_{t-1}^{(\star)} = \text{DDIM}\left(\mathbf{x}_t^{(A)}, \boldsymbol{\epsilon}_t^{(\star)}, t\right)\end{array}\right.$ | |
| Cross Attn. Manip. | $\left\{\mathbf{x}_t^{(A)} = \text{Gen}(\mathbf{x}_T, \mathbf{c}^{(A)}, t)\right.$ | $\boldsymbol{\epsilon}_t^{(\star)} = \mathcal{M}\left(\boldsymbol{\epsilon}_\theta(\mathbf{x}_t^{(A)}, \cdot, t), \mathbf{c}^{(A)}, \mathbf{c}^{(B)}, t\right)$ | $\left\{\mathbf{x}_{t-1}^{(\star)} = \text{DDIM}\left(\mathbf{x}_t^{(A)}, \boldsymbol{\epsilon}_t^{(\star)}, t\right)\right.$ | Hertz et al. (2022) Chefer et al. (2023) |
| Guidance | $\left\{\mathbf{x}_T \sim \mathcal{N}(\mathbf{0}, \mathbf{I})\right.$ | $\left\{\begin{array}{l}\boldsymbol{\epsilon}_t^{A\star} = \boldsymbol{\epsilon}_\theta\left(\mathbf{x}_t^{(\star)}, \mathbf{c}^{(A)}, t\right) \\ \boldsymbol{\epsilon}_t^{B\star} = \boldsymbol{\epsilon}_\theta\left(\mathbf{x}_t^{(\star)}, \mathbf{c}^{(B)}, t\right) \\ \boldsymbol{\epsilon}_t^{\star} = (1+\beta)\cdot\boldsymbol{\epsilon}_t^{A\star} - \beta\cdot\boldsymbol{\epsilon}_t^{B\star}\end{array}\right.$ | $\left\{\mathbf{x}_{t-1}^{(\star)} = \text{DDIM}\left(\mathbf{x}_t^{(\star)}, \boldsymbol{\epsilon}_t^{(\star)}, t\right)\right.$ | Ho & Salimans (2022) |
| Pred. Noise Interp. Pred. Noise Masking | $\left\{\{(\mathbf{x}_t^{(A)}, \boldsymbol{\epsilon}_t^{(A)})\} = \text{GenPath}\left(\mathbf{x}_T, \mathbf{c}^{(A)}, t\right)\right.$ | $\boldsymbol{\epsilon}_t^{(\star)} = \omega\boldsymbol{\epsilon}_t^{(A)} + (1-\omega)\boldsymbol{\epsilon}_t^{(\star)},$ $\boldsymbol{\epsilon}_t^{(\star)} = \mathbf{M}\odot\boldsymbol{\epsilon}_t^{(A)} + (1-\mathbf{M})\odot\boldsymbol{\epsilon}_t^{(\star)}$ | $\left\{\mathbf{x}_{t-1}^{(\star)} = \text{DDIM}\left(\mathbf{x}_t^{(A)}, \boldsymbol{\epsilon}_t^{(\star)}, t\right)\right.$ | |

Simply switching to another condition during denoising is a very straightforward way to add new semantics to the input image when doing text-guided image editing. We perform a simple experiment in Fig. 2 to illustrate that the early timesteps in the diffusion process contribute to the layout, while the later timesteps are adding semantics and details to the image. Our goal is to systematically analyze the design space for manipulating the diffusion path over time to perform these types of edits. As result of our analysis, we propose **MDP** as generalized framework for text-guided image editing and show how concurrent and previous methods fit our framework. A summary can be found in Tab. 1.

All manipulations in our framework are time-dependent and have parameters, e.g. interpolation parameters. MDP allows the user to specify different types of manipulations and different manipulation parameters at each timestep, which we call the manipulation schedule.

Assume we have an image $\mathbf{x}_0^{(A)}$ as input, along with a condition $\mathbf{c}^{(A)}$ that is used to generate $\mathbf{x}_0^{(A)}$ and a new condition $\mathbf{c}^{(B)}$. We first invert the image $\mathbf{x}_0^{(A)}$ to get an initial noise $\mathbf{x}_T$. We identify the following types of manipulations:

**Intermediate denoised output.** Starting from $\mathbf{x}_T$, we can generate image $\mathbf{x}_0^{(B)}$ using the new condition $\mathbf{c}_0^{(B)}$. Assume we have all the intermediate latents for the generation of $\mathbf{x}_t^{(A)}$, $\left\{\left(\mathbf{x}_t^{(A)}\right)\right\}_{t=[T,\dots,0]} = $ GenPath $\left(\mathbf{x}_T, \mathbf{c}^{(A)}, t\right)$. We denote the intermediate latents in the new path as $\mathbf{x}_t^{(\star)}$. We can modify the intermediate outputs $\mathbf{x}_t^{(A)}$ and $\mathbf{x}_t^{(\star)}$ with either linear interpolation or masking. We can control the resulting image by changing the interpolation factor $\omega$ (or the binary mask $\mathbf{M}$),

$$\mathbf{x}_t^{(\star)} = \mathbf{x}_t^{(A)} + (1-\omega)\mathbf{x}_t^{(\star)}, \tag{1}$$

or

$$\mathbf{x}_t^{(\star)} = \mathbf{M}\odot\mathbf{x}_t^{(A)} + (1-\mathbf{M})\odot\mathbf{x}_t^{(\star)}. \tag{2}$$

When we start this manipulation, $\mathbf{x}_t^{(\star)} = \mathbf{x}_t^{(B)}$. The strategy is used in MagicMix Liew et al. (2022) and DiffEdit Couairon et al. (2022).

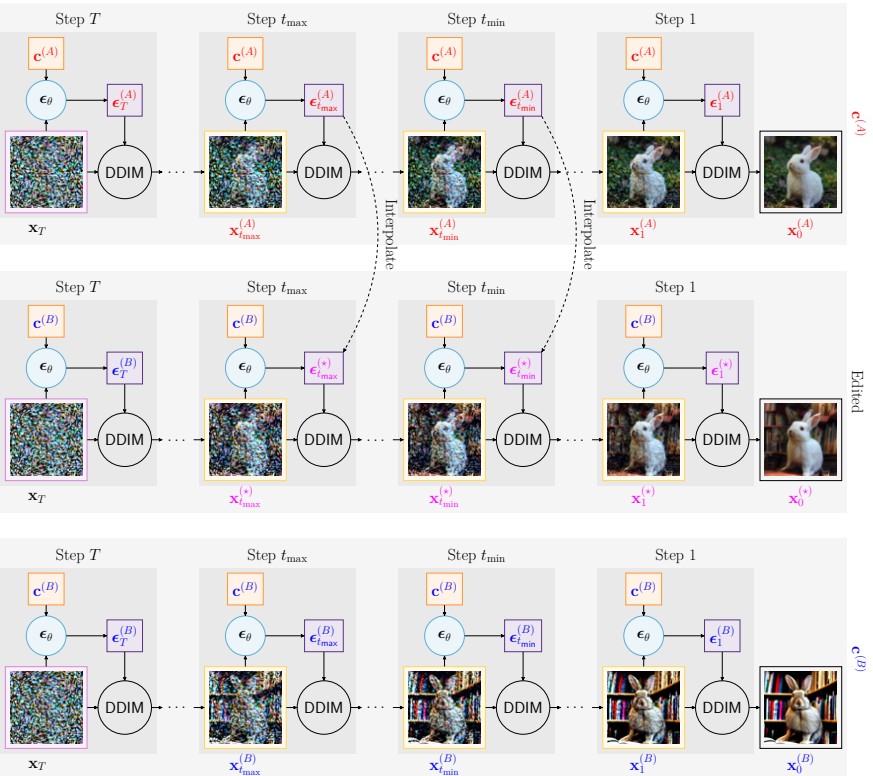

Figure 3: Predicted Noise Manipulation. The top branch is inverted from a real image given condition $\mathbf{c}^{(A)}$ "*Photo of a rabbit on the grass*". The bottom branch is generated using condition $\mathbf{c}^{(B)}$ "*Photo of a rabbit in a library*". We copy the predicted noise from step $t_{max}$ to $t_{min}$ of the top branch, then use $\mathbf{c}^{(B)}$ to denoise and generate the middle images.

**Condition embeddings.** Given two conditions $\mathbf{c}^{(A)}$ and $\mathbf{c}^{(B)}$, we can change $\mathbf{c}^{(A)}$ at timestep $t$,

$$\mathbf{c}_t^{(\star)} = \omega \mathbf{c}^{(A)} + (1 - \omega)\mathbf{c}^{(B)}, \tag{3}$$

where $\mathbf{c}_t^{(\star)}$ will be used as the new condition that embeds the information from both condition $\mathbf{c}^{(A)}$ and $\mathbf{c}^{(B)}$.

**Cross attention.** Recent condition diffusion models inject condition information $\mathbf{c}$ by cross attending between $\mathbf{c}$ and image features. Prompt-to-Prompt Hertz et al. (2022) and Attend-and-Excite Chefer et al. (2023) find that modifying the cross attention maps can give interesting image editing effects. Here, we do not dig into details of how to manipulate the cross-attention maps. Instead, we simply write the manipulation in functional form $\mathcal{M}$,

$$\boldsymbol{\epsilon}_t^{(\star)} = \mathcal{M}\left(\boldsymbol{\epsilon}_\theta\left(\mathbf{x}^{(A)}, \cdot, t\right), \mathbf{c}^{(A)}, \mathbf{c}^{(B)}\right). \tag{4}$$

**Guidance.** Classifier-free guidance Ho & Salimans (2022) proposed

$$\boldsymbol{\epsilon}_t^{(\star)} = \boldsymbol{\epsilon}_\theta\left(\mathbf{x}_t, \mathbf{c}^{(A)}, t\right) + \beta\left(\boldsymbol{\epsilon}_\theta\left(\mathbf{x}_t, \mathbf{c}^{(A)}, t\right) - \boldsymbol{\epsilon}_\theta\left(\mathbf{x}_t, \varnothing, t\right)\right), \tag{5}$$

where $\beta$ is a real number and $\varnothing$ denotes an empty condition. The $\beta$ is often called "guidance scale". The term $\boldsymbol{\epsilon}_\theta\left(\mathbf{x}_t, \varnothing, t\right)$ can be seen as unconditional output of $\boldsymbol{\epsilon}_\theta$. Similar ideas are also used in Bansal et al. (2023); Liu et al. (2023); Brack et al. (2022) for image editing. This can be generalized to

$$\boldsymbol{\epsilon}_t^{(\star)} = \boldsymbol{\epsilon}_\theta\left(\mathbf{x}_t, \mathbf{c}^{(A)}, t\right) + \beta\left(\boldsymbol{\epsilon}_\theta\left(\mathbf{x}_t, \mathbf{c}^{(A)}, t\right) - \boldsymbol{\epsilon}_\theta\left(\mathbf{x}_t, \mathbf{c}^{(B)}, t\right)\right). \tag{6}$$

Intuitively, with equation 14, we want the output to have more characteristics from $\mathbf{c}^{(A)}$ when generating using the new condition $\mathbf{c}^{(B)}$ to preserve the layout. If $\beta$ is a number between $[-1, 0]$, we obtain this linear interpolation,

$$\boldsymbol{\epsilon}_t^{(\star)} = \omega \cdot \boldsymbol{\epsilon}_\theta\left(\mathbf{x}_t, \mathbf{c}^{(A)}, t\right) + (1 - \omega) \cdot \boldsymbol{\epsilon}_\theta\left(\mathbf{x}_t, \mathbf{c}^{(B)}, t\right). \tag{7}$$

**Predicted noise.** Both the equation 4 and equation 14 are modifying predicted noise $\boldsymbol{\epsilon}_t$. Inspired by this, we investigate another manipulation. In timestep $t$, we interpolate the predicted noise $\boldsymbol{\epsilon}_t^{(A)}$ with the predicted noise that is using condition $\mathbf{c}^{(B)}$. Assume we have all the predicted noises for the generation of $\mathbf{x}_t^{(A)}$:

$$\left\{\left(\mathbf{x}_t^{(A)}, \boldsymbol{\epsilon}_t^{(A)}\right)\right\}_{t=[T,\dots,0]} = \text{GenPath}\left(\mathbf{x}_T, \mathbf{c}^{(A)}, t\right), \tag{8}$$

we can mix the two predicted noises,

$$\boldsymbol{\epsilon}_t^{(\star)} = \omega \cdot \boldsymbol{\epsilon}_\theta\left(\mathbf{x}_t^{(A)}, \mathbf{c}^{(A)}, t\right) + (1 - \omega) \cdot \boldsymbol{\epsilon}_\theta\left(\mathbf{x}_t^{(\star)}, \mathbf{c}^{(B)}, t\right), \tag{9}$$

where when we start this manipulation, $\mathbf{x}_t^{(\star)} = \mathbf{x}_t^{(B)}$. We show the pipeline of this method in Fig. 3.

## 4 Experiments and Analysis

Table 2: Quantitative results of MDP-$\boldsymbol{\epsilon}_t$ compared with baseline methods.

|  | MDP-$x_t$ | MDP-$c$ | P2P | MDP-$\beta$ | DiffEdit | MDP-$\boldsymbol{\epsilon}_t$ |
|---|---|---|---|---|---|---|
| LPIPS↓ | 0.3813 | 0.4111 | 0.2900 | 0.4528 | **0.2348** | 0.3484 |
| CLIP score↑ | 27.5084 | **28.4560** | 28.2487 | 27.9109 | 27.6915 | 28.3228 |
| CLIP directional similarity↑ | 0.2262 | 0.2062 | 0.1938 | 0.1935 | 0.1851 | **0.2265** |

### 4.1 Settings

**Editing tasks.** We describe a small taxonomy for image editing applications that we use to analyze our framework and compare it to previous work. We divide all common image editing operations into two categories: local editing and global editing. We provide the details in the Supplementary Materials.

**Our manipulations and baselines.** Given a real input image $\mathbf{x}_0^{(A)}$, we first use Null-text Inversion Mokady et al. (2022) together with a text-condition $\mathbf{c}^{(A)}$ to obtain an initial noise tensor $\mathbf{x}_T$. The initial condition $\mathbf{c}^{(A)}$ can be either an empty text, a manual prompt provided by the user or an caption generated by image captioning tools. Another given condition $\mathbf{c}^{(B)}$ that is used to guide the editing process would induce the generation of output image $\mathbf{x}_0^{(B)}$ starting from the same initial noise tensor $\mathbf{x}_T$. As we do not use a mask as an input, we only consider linear operations that do not involve a mask:

- MDP-$x_t$: intermediate latent interpolation. We linearly interpolate the intermediate latents $\mathbf{x}_t^{(A)}$ and $\mathbf{x}_t^{(B)}$ using equation 1. This manipulation differs from MagicMix Liew et al. (2022) by using Null-text Inversion to obtain the initial noise tensor $\mathbf{x}_T$ rather than directly adding noise.

- MDP-$c$: conditional embedding interpolation. We interpolate the condition $\mathbf{c}^{(A)}$ and $\mathbf{c}^{(B)}$ by using equation 3.

- P2P (Prompt-to-Prompt) Hertz et al. (2022): cross attention manipulation. P2P manipulates the cross attention maps and self attention maps according to the changes in the new text prompt compared to the input prompt.

- MDP-$\beta$: guidance. We use condition $\mathbf{c}^{(A)}$ as a guidance to inject the layout while generating new semantics by using equation 15.

- MDP-$\epsilon_t$: predicted noise interpolation. We mix the predicted noises when generating input image $\mathbf{x}_0^{(A)}$ with the noises using condition $\mathbf{c}^{(B)}$ by following equation 9.

- DiffEdit Couairon et al. (2022): intermediate latent interpolation. We also adopt DiffEdit as another competitive baseline for more comparisons.

Comparing MDP-$x_t$, MDP-$\epsilon_t$ and P2P from their formulations, MDP-$x_t$ directly injects the intermediate latent from the original image, which has the largest degree of preserving the original layout. As for MDP-$\epsilon_t$, which injects the predicted noise and can only partly affect the new intermediate latent; while for P2P, it manipulates the attention maps which can only partly affect the new predicted noise. Therefore, P2P has the weakest degree of preserving the original layout. Due to the space limit, we show the qualitative results of DiffEdit in the Supplementary Materials.

**Manipulation schedule.** We investigate how the methods work under a simple schedule using default settings. We start the manipulation at step $t_{max}$ and end at step $t_{min}$. The total number of timesteps of a manipulation is denoted as $T_M = t_{max} - t_{min}$. We perform edits only during these $T_M$ steps. For the remaining steps, we simply use $\mathbf{c}^{(B)}$ to denoise the noisy latent. We vary $t_{max}$ and $t_{min}$ and manually select the best result for each method. We fix the interpolation factors (guiding scale for MDP-$\beta$) for the following concerns: (1) We observe that setting the manipulation schedule and interpolation factor can both adjust the degree to which the layout is maintained in the edited image to some extent; (2) We want to find a good default setting of interpolation factors for each manipulation; (3) When setting the interpolation factor $\beta = 0$ for MDP-$\beta$, this manipulation is equal to MDP-$c$, which we want to avoid in order to show the characteristic of each manipulation. We therefore empirically fix the interpolation factor of MDP-$x_t$, MDP-$c$, P2P, MDP-$\beta$ and MDP-$\epsilon_t$ to be 0.7, 1, 1, -0.3, and 1, respectively. In general, $t_{max}$ is ranging from 0 to 5 while $T_M$ is set to be around 20 for each manipulation. We tune the manipulation schedule for each manipulation to obtain the desired editing result. We analyze and summarize how the methods work under various manipulation schedules in the Supplementary Materials.

**Implementation.** For text-guided editing, we test all the methods using the publicly available latent diffusion model Stable Diffusion [1]. For class-guided editing, we use the conditional latent diffusion model Rombach et al. (2022) trained on ImageNet Deng et al. (2009). We test our manipulations on one NVIDIA A100 GPU. As no training and finetuning is required, each manipulation can be generally done within 10 seconds. As the inversion method we use is built on top of the DDIM sampler Mokady et al. (2022), we also use DDIM sampler during the sampling process. However, our method can adopt other deterministic samplers.

**Evaluation.** We provide both quantitative and qualitative comparison for evaluation. For quantitative evaluation, We adopt three metrics to evaluate the results. We collected around 90 real-world images and grouped them into different editing applications for evaluation. Part of the images examples are shown in Fig. 4 and Supplementary Materials. We adopt LPIPS Zhang et al. (2018), CLIP score Hessel et al. (2022) and CLIP directional similarity Gal et al. (2021) for evaluation. Explanations of these metrics can be found in the Supplementary Materials. Despite from these metrics, which do not necessarily align with the human perspective, we conducted a user study to compare MDP-$\epsilon_t$ against baseline Prompt-to-Prompt using figures shown in the Supplementary Materials. We conducted the study on MTurk and asked the participants to evaluate based on three different aspects:

- Image layout preservation: this metric is to evaluate if the edited image preserve the overall layout of the input image.

- Image quality: this is to see if the visual quality of the edited image is good or not.

- Image-text alignment: this metric is to evaluate if the edited image applied the prompt appropriately.

---

[1]https://huggingface.co/CompVis/stable-diffusion-v1-4

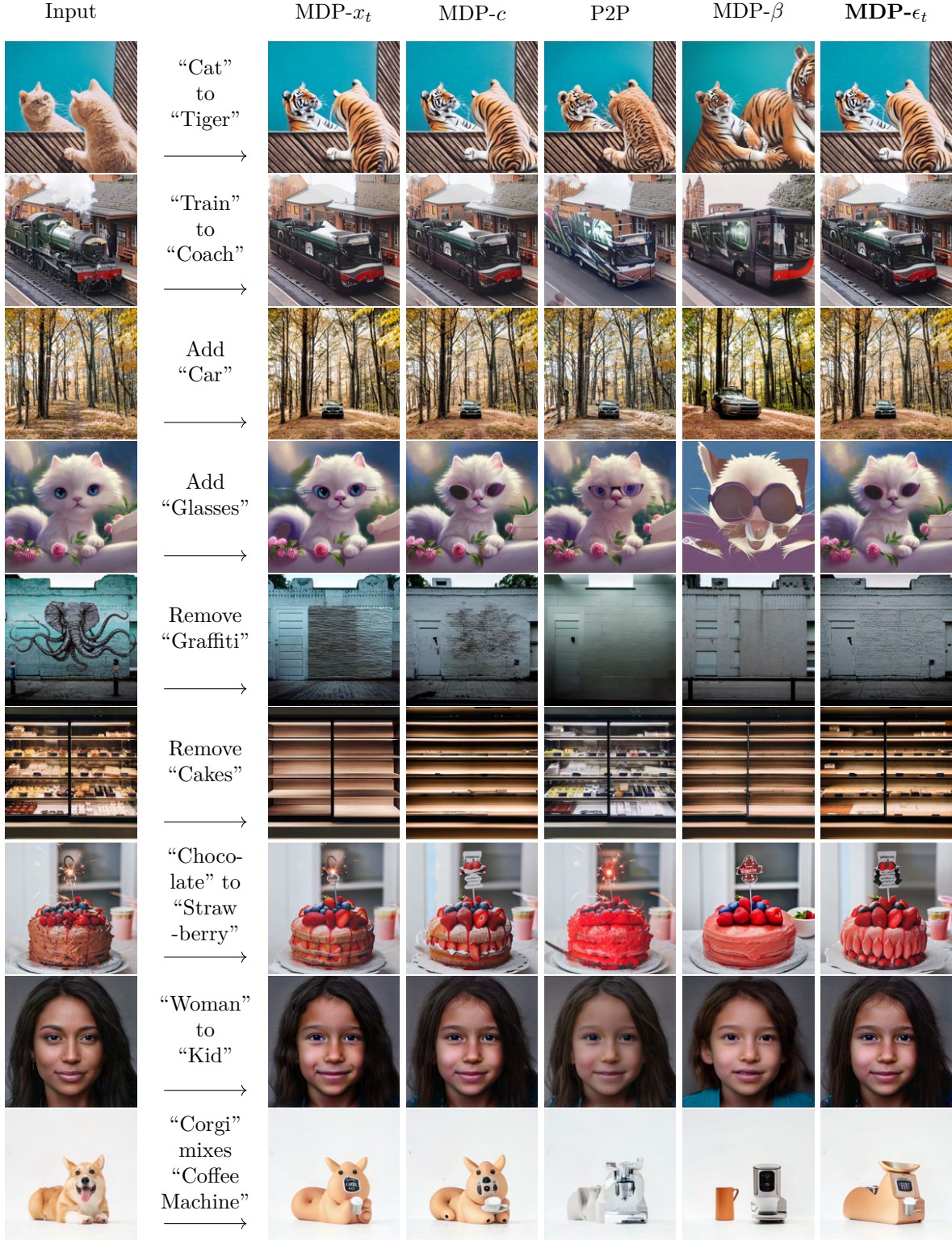

Figure 4: Text-guided local editing results.

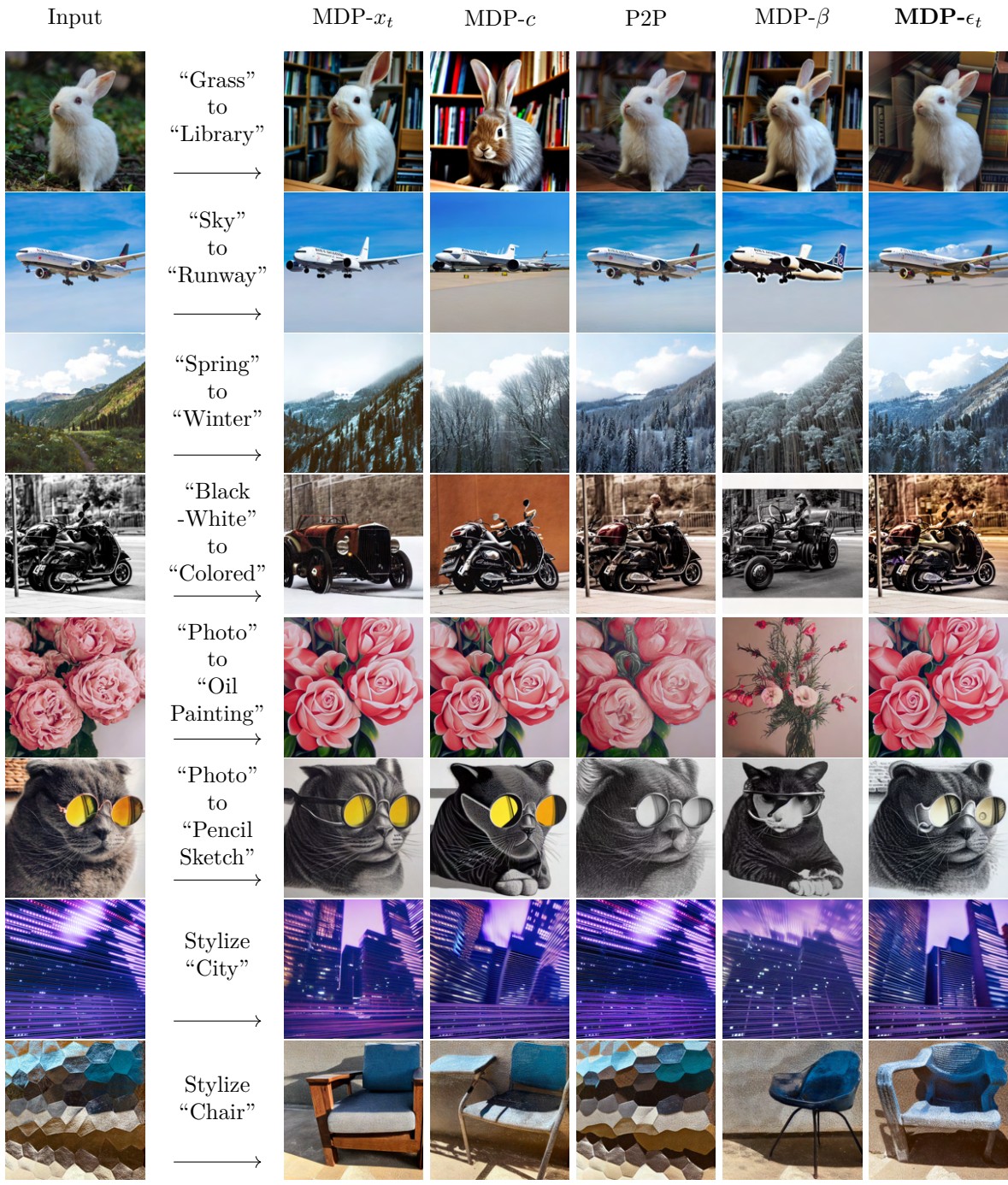

Figure 5: Text-guided global editing results.

## 4.2 Quantitative results

We show the quantitative results in Tab. 2. Results show that MDP-$\epsilon_t$ has the best consistency regarding the changes between the input image and edited images and the changes between the input condition and new condition. Although DiffEdit achieved the best LPIPS score, we argue that DiffEdit is an local editing method that only edits a localized region of an image. Therefore, the degree of changes in the edited images is usually lower. However, DiffEdit can fall short of global editing applications. We further argue that a higher number of LPIPS or CLIP score alone does not necessarily mean a better quality, as identical images can obtain almost infinite LPIPS, while images that are edited too much can also obtain very high CLIP score. Nevertheless, CLIP directional similarity, which takes input image, edited image and text prompt into consideration at the same time, plays a better role for accessing image-text alignment as well as preserving original content. In this case, MDP-$\epsilon_t$ demonstrated the best quantitative evaluation results.

As those quantitative metrics cannot fully align with the human judgement, we also show the user study results. In total, we could acquire 60 workers from MTurk to participate in our study. In detail, the edited images generated by MDP-$\epsilon_t$ were preferred over P2P in **60.6 %** of the cases w.r.t layout preservation of the input image, in **67.9 %** of the cases w.r.t. general quality of the output and in **73.0 %** of the cases w.r.t image-text alignment. We show more details of the user study in the Supplementary Materials.

## 4.3 Qualitative results

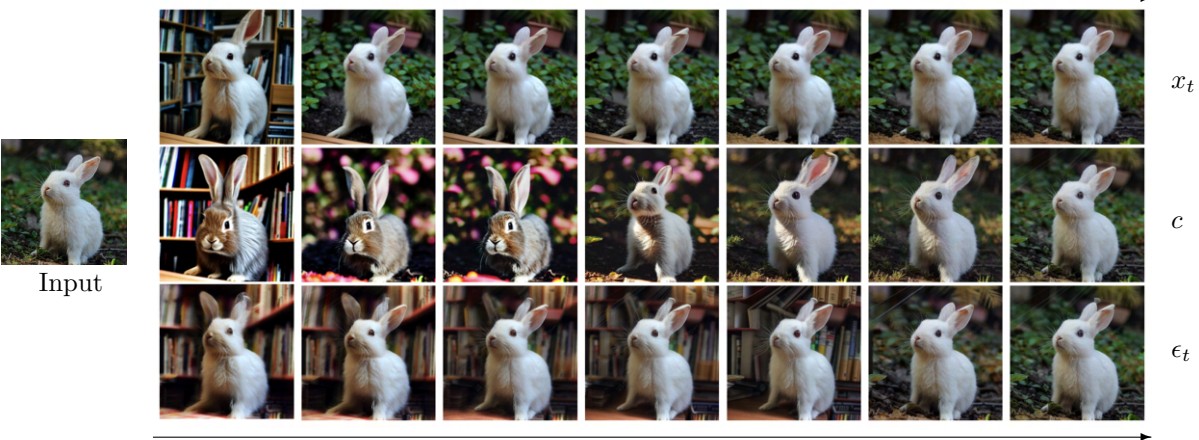

Manipulation starts from early to late

Figure 6: We change the background of the input image from "grass" to "library". We show the results of MDP-$x_t$, MDP-$c$, and MDP-$\epsilon_t$ from top to bottom. The manipulation range $T_M$ is set to be 10, 20, and 20 respectively, which are the best settings for each manipulation. Manipulation $\epsilon_t$ can do faithful edits across different manipulation ranges while the other two methods fail.

### 4.3.1 Local editing

We show examples of local edits for changing object, adding object, removing object, changing attribute, and mixing objects in Fig. 4. The edits by MDP-$\beta$ are somewhat reasonable, but the overall layout is not well preserved. In general, all the other manipulations can do the edits guided by the text prompt while preserving the background of the input image. As the initial diffusion generation steps contribute to the layout of the generated image, we thus recommend to start the manipulation at the early diffusion stage, usually $t_s$ can range from 50 to 45, and $T_M$ can be ranging from 15 to 25. For MDP-$x_t$ and MDP-$\epsilon_t$, editing can start at later steps, i.e. $t_{max}$ can be chosen to be smaller, as more layout is preserved during editing. Additionally, we observe some interesting findings that Prompt-to-Prompt fails for edits that remove objects. For the application of mixing objects, as there is no universal standard of what the mixed object should look like, we provide more examples generated by each method in the Supplementary Materials.

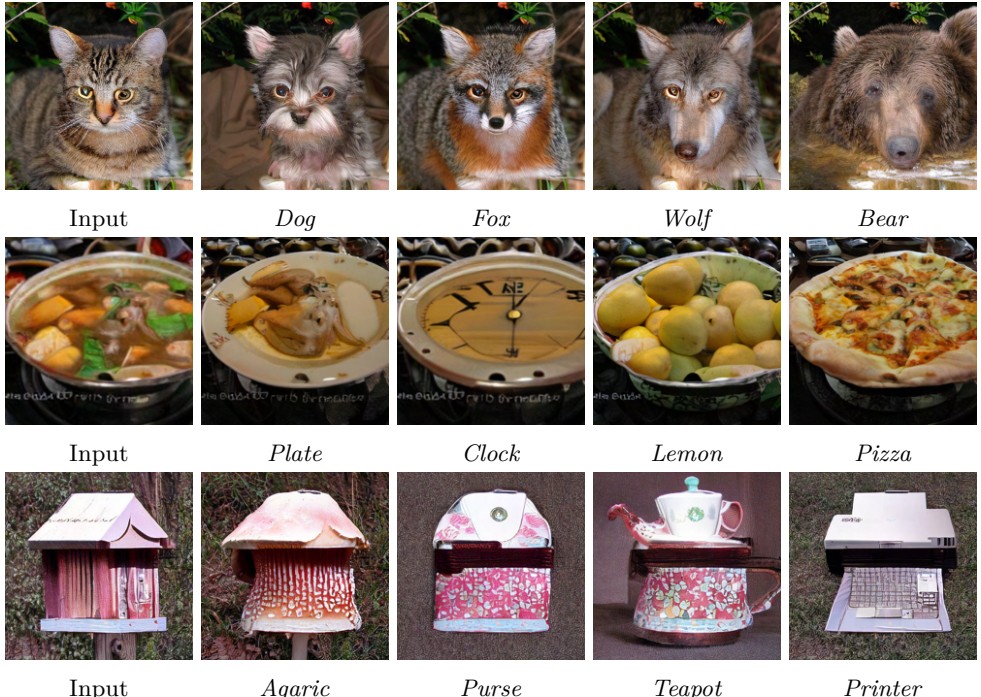

| Input | *Dog* | *Fox* | *Wolf* | *Bear* |
| Input | *Plate* | *Clock* | *Lemon* | *Pizza* |
| Input | *Agaric* | *Purse* | *Teapot* | *Printer* |

Figure 7: Class-guided image editing. We synthesize the input image using one class label, then we use another category from ImageNet to edit the input image. We use MDP-$\epsilon_t$ to demonstrate the ability to do the class-guided editing.

### 4.3.2 Global editing

We visualize the results in Fig. 5. While in local editing MDP-$x_t$ and MDP-$c$ can yield good results, in global editing they are either not able to do the edits, or the overall layout is changed too much. In contrast, We observe that MDP-$\epsilon_t$ can produce very good results most of the time even when the strong baseline method Prompt-to-Prompt (P2P) fails. Further, we compare the editing results for MDP-$x_t$ MDP-$c$ and MDP-$\epsilon_t$ under different ranges to inject layout from the input image in Fig. 6. We observe that MDP-$\epsilon_t$ can create meaningful edits when the manipulation starts at early or later stages. We therefore strongly recommend using MDP-$\epsilon_t$ when doing global editing, as it is more stable and can provide a variety of meaningful results. Here we conjecture the reason why P2P fails to perform some edits or the edited images are lack of fine-grained details. As we analyze, injecting original attention maps has a weaker effect of preserving original layout. Therefore, P2P needs to inject attention maps for more sampling timesteps to preserve layout. However, as we show in Fig. 2, semantics are formed in the later sampling timesteps, more original semantics are also injected. That means in the failure cases of P2P, it either cannot preserve the original layout, or too much original semantics are leaking to the edited images. As for DiffEdit, as it is a mask-based method, it does not perform well in the global editing applications compared to the local editing applications. We also provide more results in the Supplementary Materials.

### 4.3.3 Class condition models

We provide class-guided editing results using a class-condition diffusion model instead of a common text-conditional model. We show results from MDP-$\epsilon_t$ in Fig. 7.

## 5 Failure cases

We identify several typical failure cases of MDP-$\boldsymbol{\epsilon}_t$ in Fig. 8 with comparisons to Prompt-to-Prompt. For the first case, when we deform the object in the input image such as changing the shape, neither of MDP-$\boldsymbol{\epsilon}_t$ nor Prompt-to-Prompt can faithfully perform the edits while preserving the layout of the input image. This

kind of failure cases of deformation is also reported in the Prompt-to-Prompt paper. That means these two algorithms are better suited for performing edits without large changes to the object; in other words, the constraint to preserve the layout from the input image is relatively big in these two algorithms, which limits their ability to perform edits with larger changes. In another case MDP-$\epsilon_t$ cannot convert the red car into a yellow one, instead, Prompt-to-Prompt can do the edit. We observe that in some cases MDP-$\epsilon_t$ struggles to transform the color of the object, especially the color that is very different from the original one. We conjecture that by keeping the predicted noises from the input image path, a lot of information about the color is also preserved.

Another common kind of failure cases is map-based image translation, such as converting a segmentation map to a photo or a sketch to a photo. All the manipulations we analyze in the design space preserve the color and texture of the input image to an extent, which is because of the nature of this kind of manipulation. However, for map-based input images, the colors are usually only used to distinguish different parts of the objects, other than the real colors of the objects have. Therefore, the map-based image editing usually involves an auxiliary network or additional training to extract the useful features in order to guide the editing (Voynov et al., 2022; Avrahami et al., 2022a).

We also report another common failure case in Fig. 9, which is a result of the imperfections in image inversion. As all the manipulations in MDP for real images rely on image inversion, when the inversion algorithm cannot invert the input image in a plausible way, the resulting edited image will most likely fail.

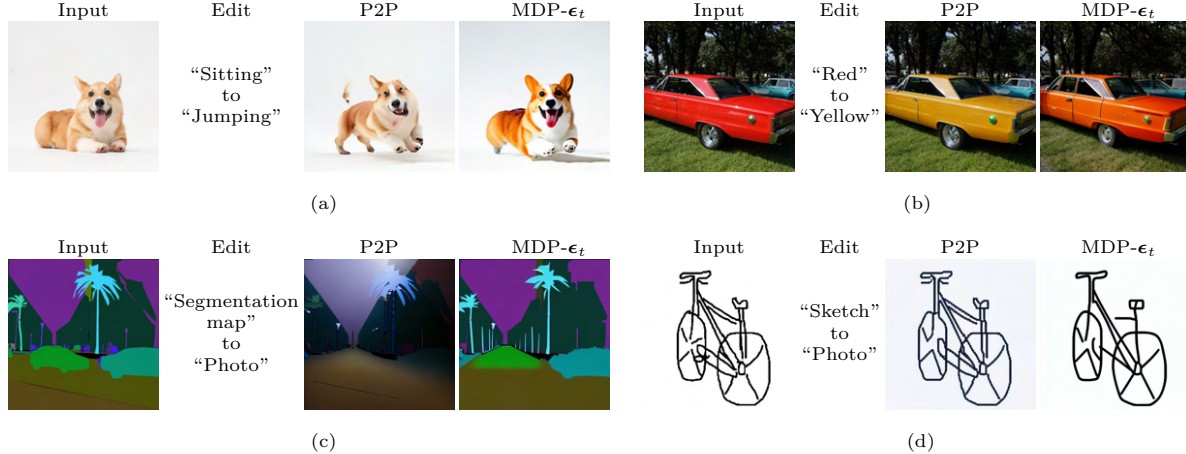

Figure 8: Typical failure cases of MDP-$\epsilon_t$. In (a) MDP-$\epsilon_t$ fails to preserve the layout of the input image; In (b) MDP-$\epsilon_t$ fails to faithfully synthesize the yellow color. In (c) and (d), MDP-$\epsilon_t$ fails to do the map-based image editing, where the color and texture of the input map has no semantic meaning.

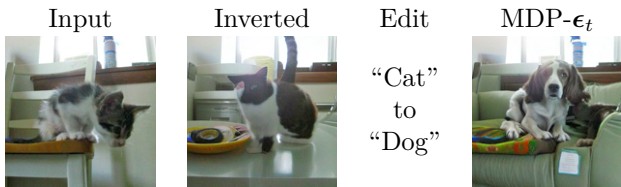

Figure 9: MDP-$\epsilon_t$ fails when the inverted image does not reconstruct the input image in a plausible way.

## 6 Conclusion and Future work

We introduce a generalized editing framework, MDP, which contains 5 different manipulations that are suitable, their parameters, and the manipulation schedule. We highlight a new manipulation by editing the predicted noise. The results show that this manipulation can achieve high quality edits in challenging cases

where previous work may fail. Our work also has multiple limitations. While we analyzed many design choices, we cannot claim that our framework is exhaustive. Some new ideas will easily be integrated into our framework while others may be fundamentally different. We did not analyze editing operations that become possible using fine-tuning or modifications of the network architecture. Also, we rely on the reconstruction ability of inversion for real images. In some cases the inversion may fail to faithfully reconstruct the input. We also consider move the discussion of our framework into a continuous space.

## 7 Broader impact

Here, we discuss several potential impacts of our work in a broader context. Our paper uses publicly available Stable Diffusion as our diffusion model, which inherits the biases from its training dataset. The biases include but are not limited to race, ethical groups, gender, socioeconomic status, and culture stereotypes. A more diverse and inclusive dataset can reduce the biases. Moreover, bias detection and correction tools are important to apply to the output of the diffusion models or be integrated into the generation process.

Diffusion models can generate harmful content, such as deepfakes, misleading images, and inappropriate material. Implementing an effective filter is crucial for detecting and screening the outputs of these models. For certain types of content, like deepfakes, which may not be easily caught by standard filters, more advanced methods should be employed to prevent misuse of the diffusion models.

Since our framework can edit images based on text prompts, there is a risk that it could be used to modify and distribute proprietary designs or artworks without permission, potentially violating copyright laws. To prevent intellectual property rights infringement, it is necessary to develop tools for verifying the originality of input images.

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

## A    More results

We show more comparisons between Prompt-to-Prompt and MDP-$\epsilon_t$ for each kind of edits we describe in the main paper. We show the edits for local editing in Fig. 21, 22, 23, 24, and 25, and for global editing in Fig. 29, 30, 31, and 32. We also show more intermediate results for mixing objects in Fig. 26. We show that in many cases, MDP-$\epsilon_t$ can perform as good as or better than Prompt-to-Prompt. Additionally, we also show editing results of MDP-$\epsilon_t$ using Stable Diffusion 2.1 [2] as the base diffusion model in Figure 27. Noted that all the input images we use here are synthesized from Stable Diffusion 2.1 instead of real images. We also provided comparisons between DiffEdit and MDP-$\epsilon_t$ in Figure 28. As an editing method that only edits local regions in the images, DiffEdit has difficulty to perform global editing applications.

## B    Taxonomy of Image Editing Applications

In this subsection we describe a small taxonomy for text-guided image editing applications that we use to analyze our framework and compare it to previous work. To be noted that this taxonomy only includes some major image editing applications but is not an exhaustive one. Given either a real or synthetic image as input, we edit the image following a condition such as a text prompt or a class label. We divide all common image editing operations into two categories: local editing and global editing. We further identify sub-categories for both local and global editing.

### B.1    Local editing

Typically, we want to perform edits while keeping the overall image layout, e.g. background or the shape of the chosen object in the input image.

- Changing object: change an object (or objects) in the image to another one. For example, if there is a basket of *apples*, we may change it to a basket of *oranges*. Also, we may change the image of a *dog* to an image of a *cat*.

- Adding object: add an object that does not exist in the original image. For example, given an image of a forest, we add a *car* in that forest. Given a cat face, we can add a pair of *sunglasses* on it.

- Removing object: remove an object from the image. For example, we can remove the eggs in a basket so that the basket becomes *empty*.

- Changing attribute: change the attribute such as color and texture of an object. Given a *red* bird, we may change it to a *blue* bird. We may change a human portrait from a *young* person to an *old* person.

- Mixing objects: combine an object in the input image with another object. The two objects we may want to mix can have different semantics, e.g. we can mix a *corgi* and a *coffee machine*, or a *chocolate bar* and a *purse*.

### B.2    Global editing

For global editing, the overall texture and style of the image can be changed while the layout and semantics should be the same.

- Changing background: change the background while keeping a foreground object untouched. For example, we change a rabbit on *grass* to a rabbit on the *moon*. We can also convert a black-and-white photo to be *colored*.

- Stylization: stylize an object or a scene using an input style image. For example, we give an *futuristic* image as a style input and we want the a generated image to have the same style.

---

[2]https://stability.ai/news/stablediffusion2-1-release7-dec-2022

- In-domain transfer: edit the input image by performing in-domain changes. For example, we transfer a photo of a valley in *summer* to a photo of a valley in *winter*, or we transfer a city during the *day* to a city at *night*.

- Out-of-domain transfer: edit the input image with out-of-domain changes. For example, we transfer a *photo* of a valley to an *oil painting* of a valley or we change a *portrait* of a human to a *cartoon* character. We can also change a *black-and-white* photo to a *colored* photo.

## C   Quantitative metrics

We adopt LPIPS Zhang et al. (2018), CLIP score Hessel et al. (2022) and CLIP directional similarity Gal et al. (2021) as our quantitative metrics:

- LPIPS: It can evaluate the distance between two images. We use LPIPS to evaluate if the edited image is similar to the input image, as to see if the edited image preserve the overall layout of the input image.

- CLIP score: It can be used to evaluate the similarity between the image-text pair. In this work we use CLIP score to evaluate if the edited image is aligned with the new condition $\mathbf{c}^{(B)}$ or not.

- CLIP directional similarity: This metric is to evaluate if the changes in images align with the changes in text. In this work we adopt this metric to see if the changes in the edited image compared to the input image align with the changes in the new condition $\mathbf{c}^{(B)}$ compared to the input condition $\mathbf{c}^{(A)}$.

## D   Preliminaries

**Denoising diffusion probabilistic models.**   To train a conditional generative diffusion models, we consider the following objective,

$$\min_{\theta} \mathbb{E}_{\mathbf{x}_0 \sim \mathcal{D}, \boldsymbol{\epsilon} \sim \mathcal{N}(\mathbf{0}, \mathbf{I}), t \sim U(1, T)} \left\| \boldsymbol{\epsilon} - \boldsymbol{\epsilon}_\theta(\mathbf{x}_t, \mathbf{c}, t) \right\|^2, \tag{10}$$

where $\mathcal{D}$ is an image dataset (could be raw image pixels or latents obtained from an image autoencoder), $\mathbf{x}_t$ is a noised version of the image $\mathbf{x}_0$, $\mathbf{c}$ is a conditional embedding (e.g., text, class label, image) and $\boldsymbol{\epsilon}_\theta(\cdot, \cdot, \cdot)$ is a neural network parameterized by $\theta$. After training, we can sample a new image given condition $\mathbf{c}$ with the commonly used DDIM sampler,

$$\begin{aligned} \mathbf{x}_{t-1} &= \mathrm{DDIM}(\mathbf{x}_t, \boldsymbol{\epsilon}_t, t) \\ &= \sqrt{\alpha_{t-1}} \cdot f_\theta(\mathbf{x}_t, \mathbf{c}, t) + \sqrt{1 - \alpha_{t-1}} \cdot \boldsymbol{\epsilon}_\theta(\mathbf{x}_t, \mathbf{c}, t), \end{aligned} \tag{11}$$

where $f_\theta(\mathbf{x}_t, \mathbf{c}, t) = \frac{\mathbf{x}_t - \sqrt{1-\alpha_t} \cdot \boldsymbol{\epsilon}_\theta(\mathbf{x}_t, \mathbf{c}, t)}{\sqrt{\alpha_t}}$, $\boldsymbol{\epsilon}_t = \boldsymbol{\epsilon}_\theta(\mathbf{x}_t, \mathbf{c}, t)$, and $\alpha_t$ is a noise schedule factor as in DDIM. The sampling is to iteratively apply the above equation by giving an initial noise $\mathbf{x}_T \sim \mathcal{N}(\mathbf{0}, \mathbf{I})$. For brevity, we only consider DDIM as our sampler.

**Image editing using pre-trained diffusion models.**   Our generalized editing framework MDP edits an image in its latent space also called noise space. Given an image $\mathbf{x}_0$ and the corresponding initial noise $\mathbf{x}_T$, we modify $\mathbf{x}_T$ or the generating process and our aim is to obtain a different $\mathbf{x}_0^\star$ which suits our needs. We summarize the symbols used in our later discussions here. One step of diffusion sampling can be represented as $\begin{cases} \boldsymbol{\epsilon}_t = \boldsymbol{\epsilon}_\theta(\mathbf{x}_t, \mathbf{c}, t), \\ \mathbf{x}_{t-1} = \mathrm{DDIM}(\mathbf{x}_t, \boldsymbol{\epsilon}_t, t). \end{cases}$   We represent the intermediate results of the generating process at timestep $t$ by $\mathbf{x}_t = \mathrm{Gen}(\mathbf{x}_T, \mathbf{c}, t)$, given an initial noise $\mathbf{x}_T$ and a condition $\mathbf{c}$. We also represent all the intermediate outputs by $\{(\mathbf{x}_t, \boldsymbol{\epsilon}_t)\}_{t=[T,\dots,0]} = \mathrm{GenPath}(\mathbf{x}_T, \mathbf{c}, t)$.

**Diffusion Inversion**  We can edit an image using its latent representation. Diffusion inversion tries to revert the generating process, i.e., given an image $\mathbf{x}_0$, to find the initial noise $\mathbf{x}_T$ (and the intermediate noised images $\mathbf{x}_t$) which generates $\mathbf{x}_0$. The first approach works by adding noise to $\mathbf{x}_0$. The noise schedule is exactly the same as the training noise schedule of DDPM. However, this process is stochastic and we cannot guarantee the obtained initial noise $\mathbf{x}_T$ can reconstruct $\mathbf{x}_0$ faithfully. The second approach is DDIM inversion based on the DDIM sampler in equation 13. We can calculate the initial noise by applying the formula iteratively, $\mathbf{x}_{t+1} = \sqrt{\alpha_{t+1}} \cdot f_\theta(\mathbf{x}_t, \mathbf{c}, t) + \sqrt{1 - \alpha_{t+1}} \cdot \boldsymbol{\epsilon}_\theta(\mathbf{x}_t, \mathbf{c}, t)$. But when working with classifier-free guidance, the reconstruction quality is not satisfying. The third inversion method is Null-text Inversion Mokady et al. (2022). It is based on the DDIM inversion and optimized for high reconstruction quality. In most cases, we will use the Null-text Inversion as our diffusion inversion method.

## E   Algorithms

We show implementation details about MDP-$\boldsymbol{\epsilon}_t$ in Alg. 1, MDP-$\mathbf{c}$ in Alg. 2, MDP-$\mathbf{x}_t$ in Alg. 3, and MDP-$\boldsymbol{\beta}$ in Alg. 4. For real image editing, we use Null-text inversion to obtain $\mathbf{x}_T$. For synthetic image editing, $\mathbf{x}_T$ is sampled from the standard Gaussian distribution.

---

**Algorithm 1** MDP-$\boldsymbol{\epsilon}_t$.

---

**Require:** $\{\omega_t\}_{t=[T,\ldots,0]}, \mathbf{x}_T, \mathbf{c}^{(A)}, \mathbf{c}^{(B)}, \left\{\left(\mathbf{x}_t^{(A)}, \boldsymbol{\epsilon}_t^{(A)}\right)\right\}_{t=[T,\ldots,0]} = \text{GenPath}\left(\mathbf{x}_T, \mathbf{c}^{(A)}, t\right)$

$\quad \mathbf{x}_T^{(\star)} = \mathbf{x}_T$
$\quad$ **for** $t$ in $\{T, T-1, \ldots, 1\}$ **do**
$\quad\quad \boldsymbol{\epsilon}_t^{(B)} = \boldsymbol{\epsilon}_\theta\left(\mathbf{x}_t^{(\star)}, \mathbf{c}^{(B)}, t\right)$
$\quad\quad \boldsymbol{\epsilon}_t^{(\star)} = (1 - \omega_t)\boldsymbol{\epsilon}_t^{(B)} + \omega_t\boldsymbol{\epsilon}_t^{(A)}$
$\quad\quad \mathbf{x}_{t-1}^{(\star)} = \text{DDIM}\left(\mathbf{x}_t^{(\star)}, \boldsymbol{\epsilon}_t^{(\star)}, t\right)$
$\quad$ **end for**

---

**Algorithm 2** MDP-$\mathbf{c}$.

---

**Require:** $\{\omega_t\}_{t=[T,\ldots,0]}, \mathbf{x}_T, \mathbf{c}^{(A)}, \mathbf{c}^{(B)}$
$\quad \mathbf{x}_T^{(\star)} = \mathbf{x}_T$
$\quad$ **for** $t$ in $\{T, T-1, \ldots, 1\}$ **do**
$\quad\quad \mathbf{c}_t^{(\star)} = (1 - \omega_t)\mathbf{c}^{(B)} + \omega_t\mathbf{c}^{(A)}$
$\quad\quad \boldsymbol{\epsilon}_t^{(\star)} = \boldsymbol{\epsilon}_\theta\left(\mathbf{x}_t^{(\star)}, \mathbf{c}_t^{(\star)}, t\right)$
$\quad\quad \mathbf{x}_{t-1}^{(\star)} = \text{DDIM}\left(\mathbf{x}_t^{(\star)}, \boldsymbol{\epsilon}_t^{(\star)}, t\right)$
$\quad$ **end for**

---

**Algorithm 3** MDP-$\mathbf{x}_t$.

---

**Require:** $\{\omega_t\}_{t=[T,\ldots,0]}, \mathbf{x}_T, \mathbf{c}^{(A)}, \mathbf{c}^{(B)}, \left\{\left(\mathbf{x}_t^{(A)}, \boldsymbol{\epsilon}_t^{(A)}\right)\right\}_{t=[T,\ldots,0]} = \text{GenPath}\left(\mathbf{x}_T, \mathbf{c}^{(A)}, t\right)$

$\quad \mathbf{x}_T^{(\star)} = \mathbf{x}_T$
$\quad$ **for** $t$ in $\{T, T-1, \ldots, 1\}$ **do**
$\quad\quad \boldsymbol{\epsilon}_t^{(B\star)} = \boldsymbol{\epsilon}_\theta\left(\mathbf{x}_t^{(\star)}, \mathbf{c}_t^{(B)}, t\right)$
$\quad\quad \mathbf{x}_{t-1}^{(B\star)} = \text{DDIM}\left(\mathbf{x}_t^{(\star)}, \boldsymbol{\epsilon}_t^{(B\star)}, t\right)$
$\quad\quad \mathbf{x}_{t-1}^{(\star)} = (1 - \omega_t)\mathbf{x}_{t-1}^{(B\star)} + \omega_t\mathbf{x}_{t-1}^{(A)}$
$\quad$ **end for**

---

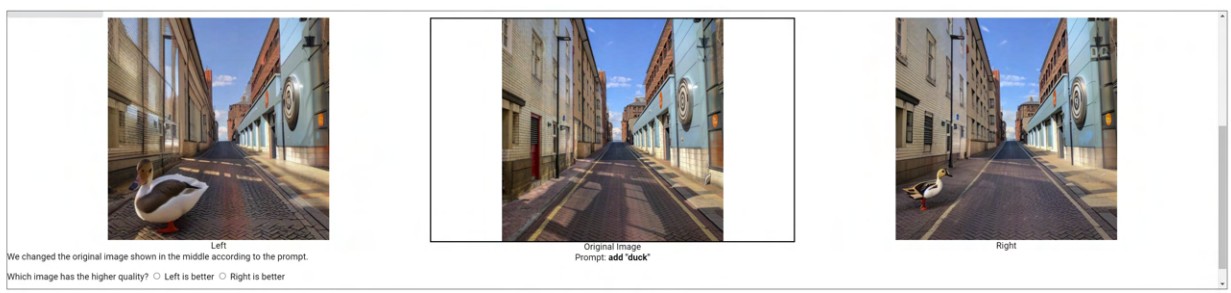

Figure 10: Screenshot of the interface for the user study.

---

**Algorithm 4** MDP-$\beta$.

---

**Require:** $\{\omega_t\}_{t=[T,\ldots,0]}, \mathbf{x}_T, \mathbf{c}^{(A)}, \mathbf{c}^{(B)}$

$\quad \mathbf{x}_T^{(\star)} = \mathbf{x}_T$

$\quad$ **for** $t$ in $\{T, T-1, \ldots, 1\}$ **do**

$\quad\quad \boldsymbol{\epsilon}_t^{(A\star)} = \boldsymbol{\epsilon}_\theta\left(\mathbf{x}_t^{(\star)}, \mathbf{c}^{(A)}, t\right)$

$\quad\quad \boldsymbol{\epsilon}_t^{(B\star)} = \boldsymbol{\epsilon}_\theta\left(\mathbf{x}_t^{(\star)}, \mathbf{c}^{(B)}, t\right)$

$\quad\quad \boldsymbol{\epsilon}_t^{(\star)} = (1-\omega_t)\boldsymbol{\epsilon}_t^{(B\star)} + \omega_t\boldsymbol{\epsilon}_t^{(A\star)}$

$\quad\quad \mathbf{x}_{t-1}^{(\star)} = \text{DDIM}\left(\mathbf{x}_t^{(\star)}, \boldsymbol{\epsilon}_t^{(\star)}, t\right)$

$\quad$ **end for**

---

## F    Availability of the initial condition

In practice, when editing real images, users need to provide $\mathbf{c}^{(A)}$ as the initial caption for the input image. This caption is used during the inversion of the real image and guides the synthesis of the edited image. A pre-trained image captioning tool, like BLIP2, can be used to generate a caption for the input image. However, the generated caption may not always be optimal for producing the best editing results. For instance, if we want to edit an image of a forest in Spring to look like Winter, including the keyword "Spring" in $\mathbf{c}^{(A)}$ can offer more precise guidance for the diffusion models to edit from "Spring" to "Winter". While it is possible to omit the keyword or even use an empty text prompt, we found that specifying the keyword explicitly often results in more stable inversion and editing outcomes. However, an image captioner might not automatically generate a caption containing this keyword. Since both the input condition $\mathbf{c}^{(A)}$ and the new condition $\mathbf{c}^{(B)}$ are crucial for achieving satisfactory edits, future work could focus on developing efficient methods for generating captions tailored to text-guided image editing tasks.

## G    User study

We show the interface of the user study presented to the participants in Figure 10. We show the input image and the editing prompt in the middle column, along with two edited images in the left and right column, respectively. One of the edited image is from MDP-$\boldsymbol{\epsilon}_t$ while another is from P2P. We randomly place either MDP-$\boldsymbol{\epsilon}_t$ on the left or right side. For each example, we ask the user three questions which correspond to the evaluation aspects we discuss in the main paper:

- Image layout preservation: *"Which image better preserves the layout of the input image?"*

- Image quality: *"Which image has the higher quality?"*

- Image-text alignment: *"Which image applies the prompt more appropriately?"*

# H   More Analysis in the Design Space

We observe that the manipulation schedule has a relatively larger influence on MDP-$\mathbf{x}_t$, MDP-$\mathbf{c}$, and MDP-$\boldsymbol{\epsilon}_t$. For MDP-$\boldsymbol{\beta}$ the linear facfor (guidance scale) has a much more important effect towards the controllability of the edited image. For Prompt-to-Prompt (a manipulation for attention maps) the range of time steps to inject the self-attention maps strongly determines what the edited image will look like. Therefore, for MDP-$\mathbf{x}_t$, MDP-$\mathbf{c}$, and MDP-$\boldsymbol{\epsilon}_t$, we explore four different manipulation schedules: constant, linear, cosine, and exponential (Fig. 11). Given an integer timestep $t \in [0, 50]$, we give the equations for the linear factor for constant schedule ($\omega_t^{\text{const}}$), linear schedule ($\omega_t^{\text{linear}}$), cosine schedule ($\omega_t^{\text{cos}}$) and exponential schedule linear schedule ($\omega_t^{\text{exp}}$):

$$
\begin{aligned}
\omega_t^{\text{const}} &= \begin{cases} 1 & \text{if } t_{\min} \leq t \leq t_{\max}, \\ 0 & \text{else.} \end{cases} \\
\omega_t^{\text{linear}} &= \begin{cases} \frac{t - t_{\min}}{50 - t_{\min}} & \text{if } t_{\min} \leq t \leq 50, \\ 0 & \text{else.} \end{cases} \\
\omega_t^{\text{cosine}} &= \begin{cases} \cos\left( \frac{\pi}{2} \frac{50 - t}{50 - t_{\min}} \right) & \text{if } t_{\min} \leq t \leq 50, \\ 0 & \text{else.} \end{cases} \\
\omega_t^{\text{exp}} &= \begin{cases} \exp\left( -5\left( \frac{50 - t}{50 - t_{\min}} \right) \right) & \text{if } t_{\min} \leq t \leq 50, \\ 0 & \text{else.} \end{cases}
\end{aligned}
\tag{12}
$$

The results are shown in Fig.12, 13, 14, 15, 16, 17. For MDP-$\boldsymbol{\beta}$, we vary the guidance scale from -0.7 to 0.7 and show the results in Fig.18. For Prompt-to-Prompt, we use the constant schedule to inject the attention maps and show the results in Fig.19 and Fig.20. We only visualize one example for global editing as we show in the paper: given an airplane image input, we change its background from "sky" to "airport runway". We only show this as an example because this example is challenging so we can identify the ability of different methods. For Prompt-to-Prompt, we show an additional example from local editing for a better understanding of the method. We discuss the results in the following subsections.

For the parameters in the design space for each manipulation, we show a general recommendation in Tab.3 and Tab.4, which is summarized based on the examples we have tried. Note that this is only considered a recommendation for general cases. For specific applications (and input images), improved results can be obtained by fine-tuning the parameters.

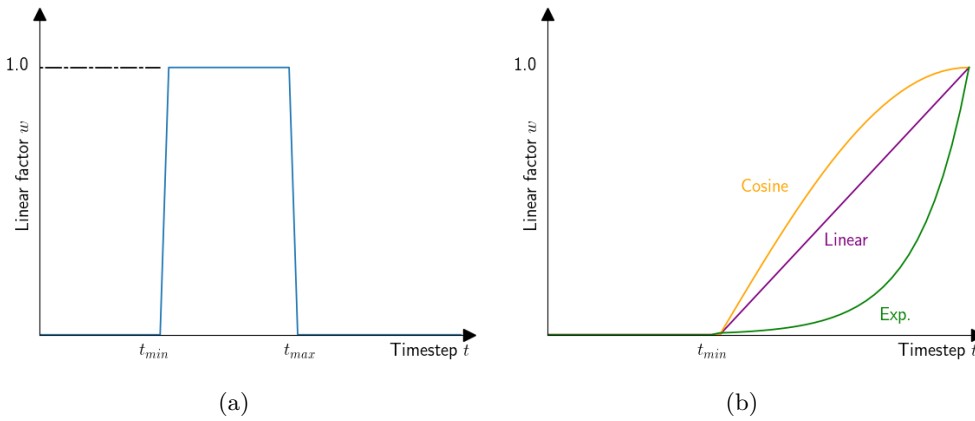

(a)                                                                (b)

Figure 11: Linear schedule is shown in (a), while linear, cosine, and exponential schedule are shown in purple, orange, and green, respectively, in (b). For linear schedule, we fix the linear factor as 1.0 while varying $t_{\max}$ and $t_{\min}$ by varying $T_M$. For the other three schedules, we fix the $t_{\max} = 50$ then vary scale factors and $t_{\min}$.

| | Suitable or not | $t_{\max}$ | $T_M$ | Manipulation schedule | Linear factor | Other |
|---|---|---|---|---|---|---|
| $x_t$ | Partly | [42, 48] | [15, 20] | Constant or others | < 1.0 | – |
| $c$ | Yes | [47, 50] | [15, 25] | Constant or others | Can be 1.0 or smaller | – |
| P2P | Yes | Usually 50 | 40 | Can be 1.0 or smaller | Can be 1.0 or smaller | The timesteps to inject self-attention maps are critical |
| $\beta$ | Partly | 50 | 50 | Does not matter | [-0.8, -0.5] | The guidance scale is very sensitive |
| $\epsilon_t$ | Yes | [44, 50] | [15, 25] | Constant or others | Can be 1.0 or smaller | – |

Table 3: The recommended parameters in the design space for local edits.

| | Suitable or not | $t_{\max}$ | $T_M$ | Manipulation schedule | Linear factor | Other |
|---|---|---|---|---|---|---|
| $x_t$ | Partly | [40, 50] | [10, 15] | Constant or others | < 1.0 | – |
| $c$ | Partly | 50 | [20, 25] | Constant or others | Can be 1.0 or smaller | – |
| P2P | Yes | Usually 50 | 40 | Constant or others | Can be 1.0 or smaller | The timesteps to inject self-attention maps are critical |
| $\beta$ | Partly | 50 | 50 | Does not matter | [-0.8, 0.5] | The guidance scale is very sensitive |
| $\epsilon_t$ | Yes | [44, 48] | [15, 25] | Constant or others | Can be 1.0 or smaller | – |

Table 4: The recommended parameters in the design space for global edits.

**MDP-$\mathbf{x}_t$, MDP-$\mathbf{c}$ and MDP-$\boldsymbol{\epsilon}_t$**   While in most of the local editing applications and some of the global editing applications, MDP-$\mathbf{x}_t$ can faithfully do the edits, for the example we show in the figures, all the examples generated by MDP-$\mathbf{x}_t$ under different schedules fail. The edited image preserves the overall layout of the input image, however, the new semantics fail to be injected. We conjecture that keeping the intermediate latent from the path when generating the input image will impose a very strong condition to preserve the information from the input image. For the result in the main paper, we fix the linear factors for MDP-$\mathbf{x}_t$ to be 0.7 under all the edits. However, for a clear comparison with other two similar methods MDP-$\mathbf{c}$ and MDP-$\boldsymbol{\epsilon}_t$, we fix the linear factors as 1.0, which results in that the new condition has no influence when $t_{\min} \leq t \leq t_{\max}$. That is also the reason why we empirically set the linear factor of MDP-$\mathbf{x}_t$ to be 0.7 for all the results in the main paper, rather than 1.0 as the other two manipulations.

On the other hand, MDP-$\mathbf{c}$ and MDP-$\boldsymbol{\epsilon}_t$ can inject the new semantics under the same schedules when linear factors are all set to be 1.0 when $t_{\max}$. For MDP-$\mathbf{c}$, typically when $t_{\max}$ is smaller than 50 or $T_M$ is smaller than 15, the new semantics can be injected into the edited image. However, the layout information from the input image is also lost. For MDP-$\boldsymbol{\epsilon}_t$, in terms of constant schedule, when $T_M$ is 25 and $t_{\max}$ is smaller than 50, the edited image can both preserve the layout and incorporate the new semantics.

Empirically, we observe that MDP-$\mathbf{x}_t$ is the strongest one to inject layout information, then is MDP-$\boldsymbol{\epsilon}_t$, while MDP-$\mathbf{c}$ is the weakest one when using the same manipulation schedule. Theoretically, this observation is aligned with the diffusion generation formula:

$$\begin{aligned}
\mathbf{x}_{t-1} &= \text{DDIM}(\mathbf{x}_t, \boldsymbol{\epsilon}_t, t) \\
&= \sqrt{\alpha_{t-1}} \cdot f_\theta(\mathbf{x}_t, \mathbf{c}, t) + \sqrt{1 - \alpha_{t-1}} \cdot \boldsymbol{\epsilon}_\theta(\mathbf{x}_t, \mathbf{c}, t),
\end{aligned} \tag{13}$$

where $f_\theta(\mathbf{x}_t, \mathbf{c}, t) = \frac{\mathbf{x}_t - \sqrt{1-\alpha_t} \cdot \boldsymbol{\epsilon}_\theta(\mathbf{x}_t, \mathbf{c}, t)}{\sqrt{\alpha_t}}$, $\boldsymbol{\epsilon}_t = \boldsymbol{\epsilon}_\theta(\mathbf{x}_t, \mathbf{c}, t)$, and $\alpha_t$ is a noise schedule factor as in DDIM. The sampling is to iteratively apply the above equation by giving an initial noise $\mathbf{x}_T \sim \mathcal{N}(\mathbf{0}, \mathbf{I})$. When setting the linear factors as 1.0, MDP-$\mathbf{x}_t$ directly replaces the intermediate latent from the path when generating the input image, while MDP-$\boldsymbol{\epsilon}_t$ replaces the predicted noise and keeps the previous intermediate latent when applying equation 13, and MDP-$\mathbf{c}$ only replaces the conditional embedding when predicting the noise.

We also observe that when $t_{\max} = 50$, i.e., the manipulation happens right at the start of the whole denoising process, and the layout information can be injected more than when $t_{\max}$ is smaller. In addition, considering

linear, cosine, and exponential schedule, for cosine schedule it can preserve the most layout formation, while the exponential schedule can preserve the least. This is also aligned with the design of the schedule, where the linear factor of the cosine schedule in every timestep is larger than which of the linear schedule and exponential schedule, which means in the cosine schedule larger amount of the layout information is taken into the consideration compared to linear and exponential schedule. These findings can also support that the initial timesteps during the generation process, especially the first denoising timestep $T_M = 50$, have a great influence on the layout of the edited image. Injecting the layout information during the early generation stages is effective.

Considering the results we have obtained, we have several conclusions:

- For the same manipulation schedule, MDP-$\mathbf{x}_t$ injects layout information the strongest from the input image, then comes MDP-$\boldsymbol{\epsilon}_t$, while MDP-$\mathbf{c}$ is the weakest one.

- For MDP-$\mathbf{x}_t$, we recommend to set the linear factor smaller than 1.0 when using the constant schedule, or consider using schedule with decreasing linear factors during the generation process, such as linear, cosine, or exponential schedules. For MDP-$\mathbf{c}$, we recommend using it only in local editing applications. For our proposed and highlighted MDP-$\boldsymbol{\epsilon}_t$, we recommend to set $t_{\max}$ to be around 44 to 50 while $T_M$ to be around 20 to 25.

- As the early stages during the diffusion generation process has a larger influence on the layout of the generated image, we recommend that for every kind of manipulation, we set the $t_{\max}$ to be larger than 44.

**MDP-$\boldsymbol{\beta}$**  Recall the formula we use for MDP-$\boldsymbol{\beta}$:

$$
\begin{aligned}
\boldsymbol{\epsilon}_t^{(\star)} =& \boldsymbol{\epsilon}_\theta \left( \mathbf{x}_t, \mathbf{c}^{(A)}, t \right) + \\
& \beta \left( \boldsymbol{\epsilon}_\theta \left( \mathbf{x}_t, \mathbf{c}^{(A)}, t \right) - \boldsymbol{\epsilon}_\theta \left( \mathbf{x}_t, \mathbf{c}^{(B)}, t \right) \right),
\end{aligned}
\tag{14}
$$

where $\beta \in \mathbf{R}$ is called guidance scale. Intuitively, with equation 14, we want the output to have more characteristics from $\mathbf{c}^{(A)}$ when generating using the new condition $\mathbf{c}^{(B)}$ to preserve the layout. This is a bit different from the original classifier free guidance Ho & Salimans (2022), where condition $\mathbf{c}^{(B)}$ is an empty condition and $\beta$ is a positive real number. In our specific image editing task, we find that when using a positive $\beta$ (e.g., the first row in Fig. 18), the edited image will be too close to condition $\mathbf{c}^{(A)}$ while losing both the layout from the input image and the new semantics from condition $\mathbf{c}^{(B)}$. Only when the guidance scale $\beta$ is a negative number, can the new semantics be added into the edited image. In this situation, the guidance becomes a linear interpolation operation:

$$
\boldsymbol{\epsilon}_t^{(\star)} = \omega \cdot \boldsymbol{\epsilon}_\theta \left( \mathbf{x}_t, \mathbf{c}^{(A)}, t \right) + (1 - \omega) \cdot \boldsymbol{\epsilon}_\theta \left( \mathbf{x}_t, \mathbf{c}^{(B)}, t \right),
\tag{15}
$$

where $\omega = 1 - \beta$ is a linear factor from 0 to 1, which means both condition $\mathbf{c}^{(A)}$ and condition $\mathbf{c}^{(B)}$ contribute positively to the generation process. Overall, $\beta$ has a larger influence over the edited image than varying the manipulation timesteps. We empirically find that $\beta \in [-0.6, -0.8]$ can obtain better results.

**Prompt-to-Prompt**  We refer to the official implementation[3], which also utilizes Null-text Inversion Mokady et al. (2022) to invert the real image and uses DDIM as the sampler. We observe that instead of replacing cross-attention maps, injecting self-attention maps has a bigger influence on the edited images. As in Fig.19 the effect of injecting self-attention maps is not obvious enough, we additionally show one local editing example in Fig.20, where when fixing the timesteps to inject self-attention maps, the edited image look almost the same under different timesteps to inject cross attention maps; while under different timesteps to inject the self-attention maps, the generated images look more different.

---

[3]https://github.com/google/prompt-to-prompt

We therefore suggest a tuning of the timesteps to inject the self-attention maps. The more timesteps to do that manipulation, the more the edited image will look like the input image. In general, for global editing, where the changes from the input image to the edited image should be larger, a smaller $T_M$ can be considered compared to which for local editing. A general setting for $T_M$ to replace the cross attention map like 20 or 40 is enough.

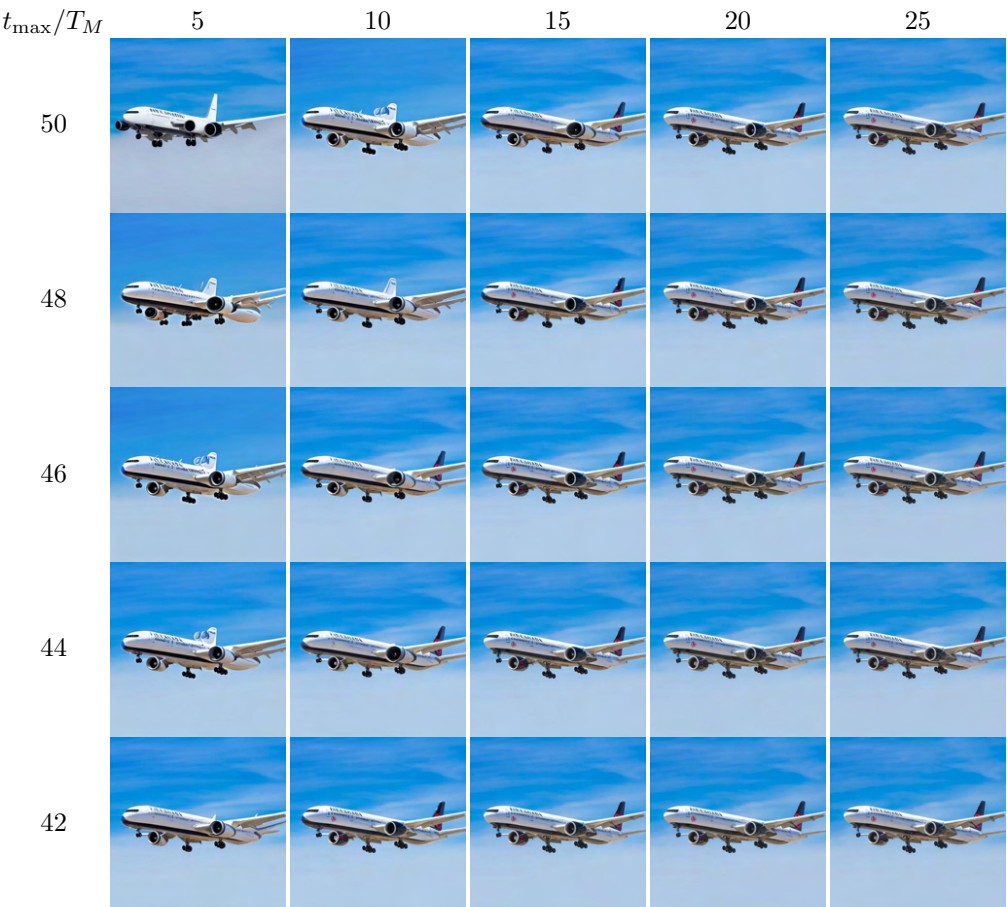

Figure 12: Results of **MDP-x**$_t$ using constant schedule.

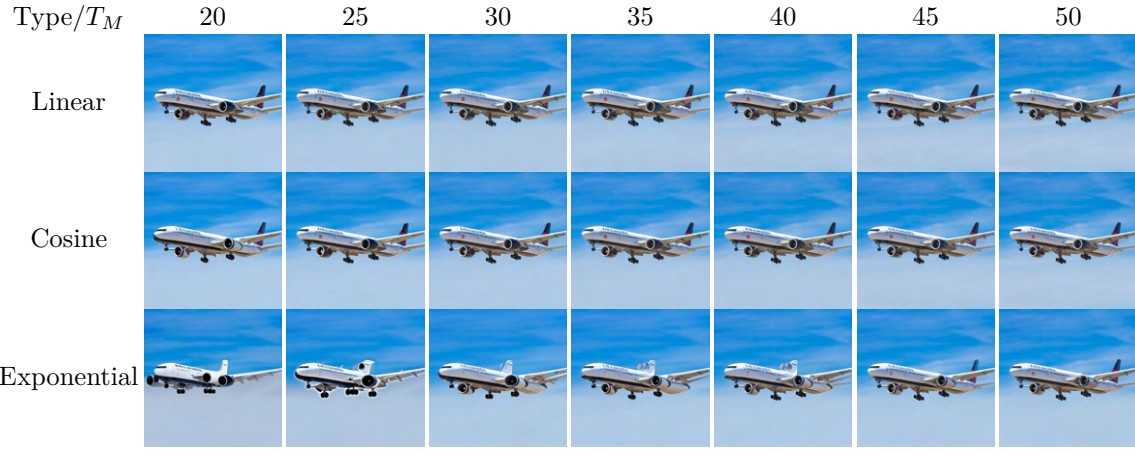

Figure 13: Results of **MDP-x**$_t$ using linear, cosine and exponential schedule.

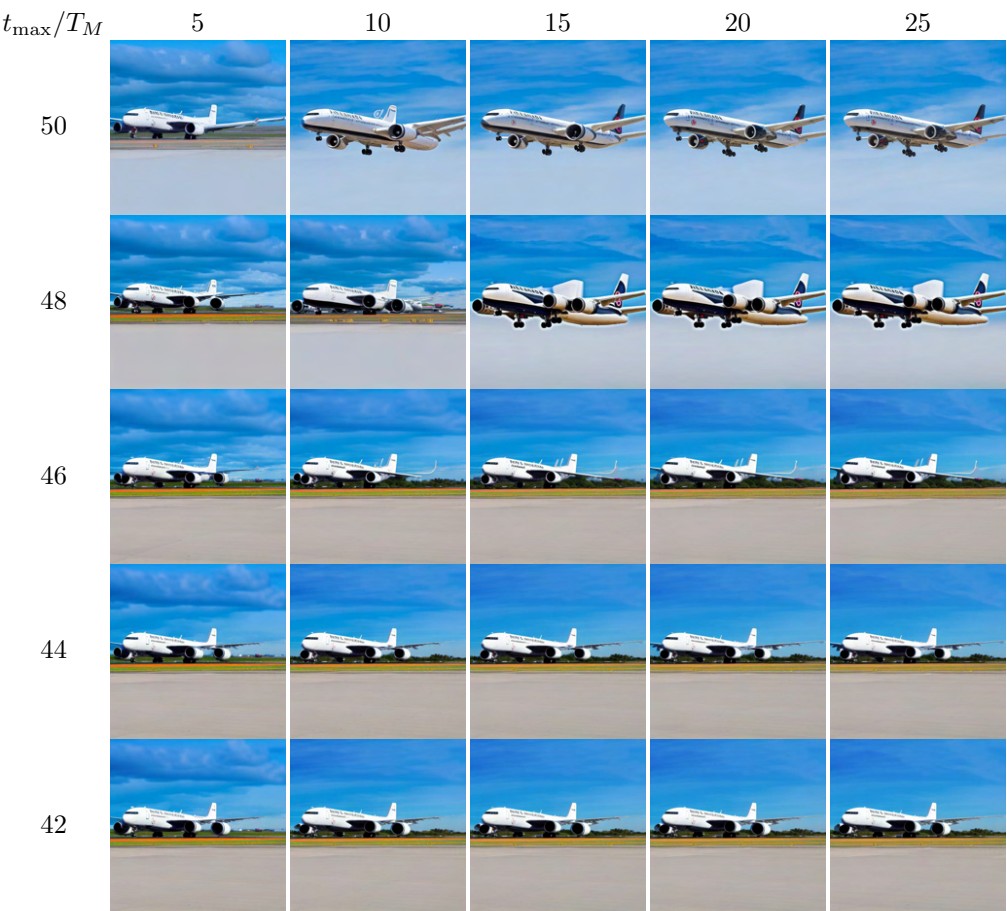

Figure 14: Results of **MDP-c** using constant schedule.

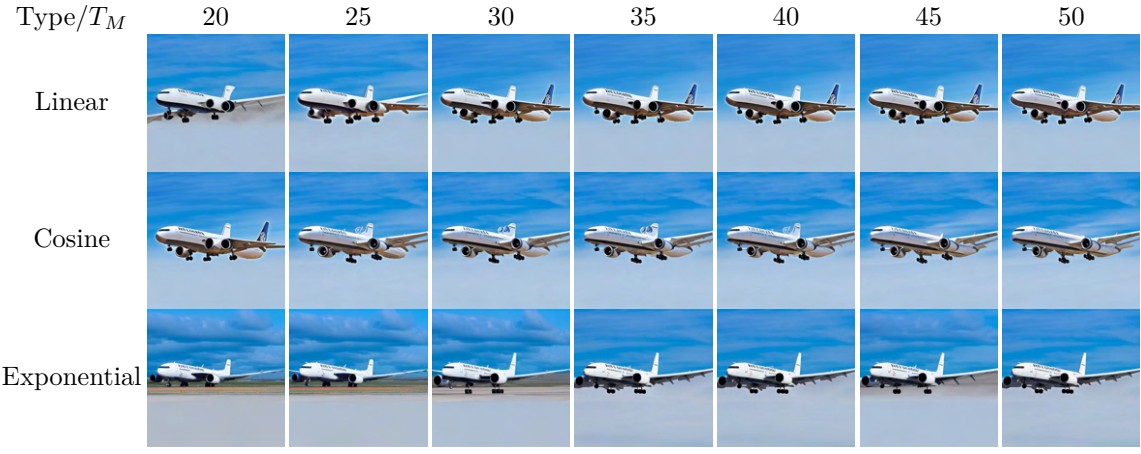

Figure 15: Results of **MDP-c** using linear, cosine and exponential schedule.

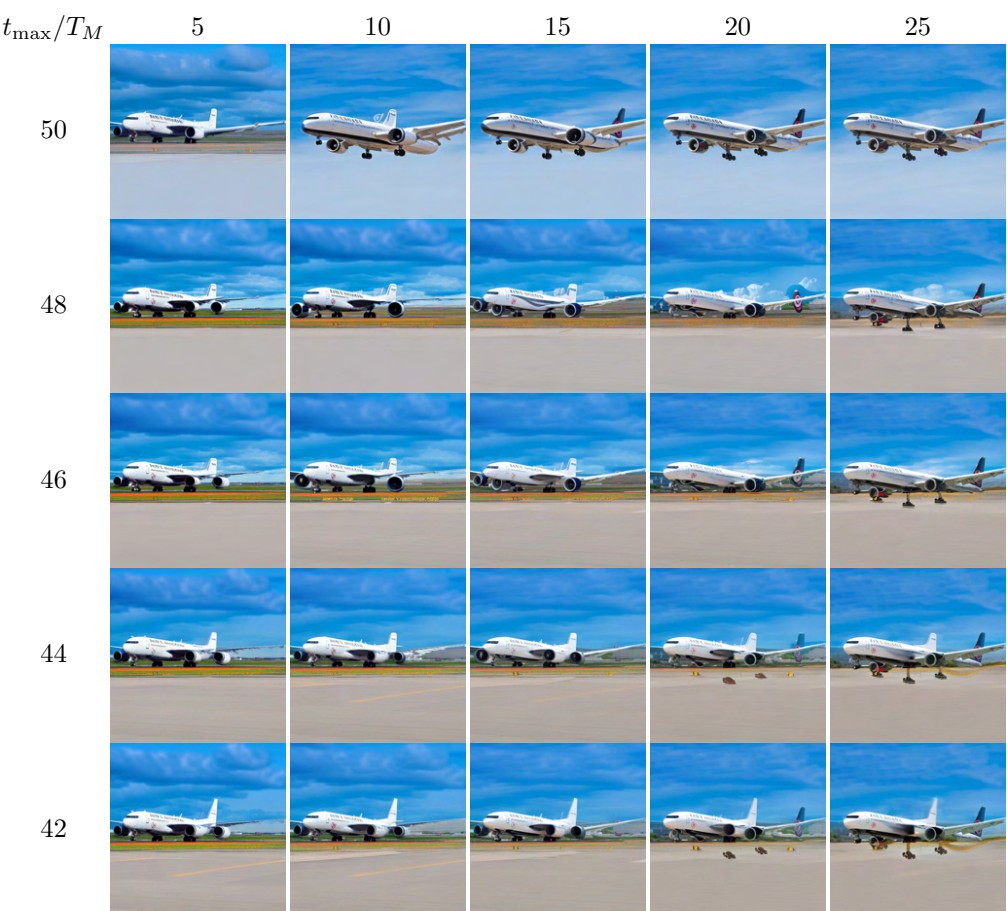

Figure 16: Results of **MDP-$\epsilon_t$** using constant schedule.

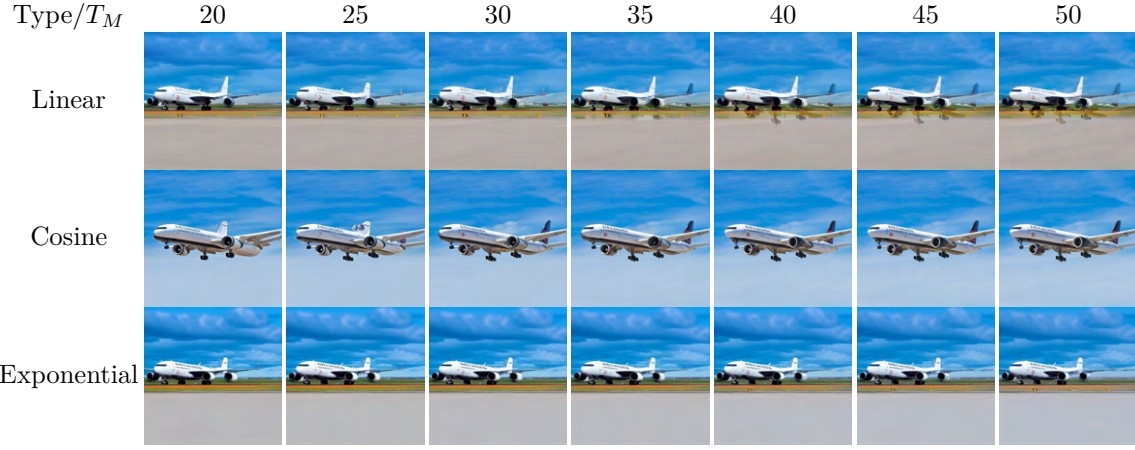

Figure 17: Results of **MDP-$\epsilon_t$** using linear, cosine and exponential schedule.

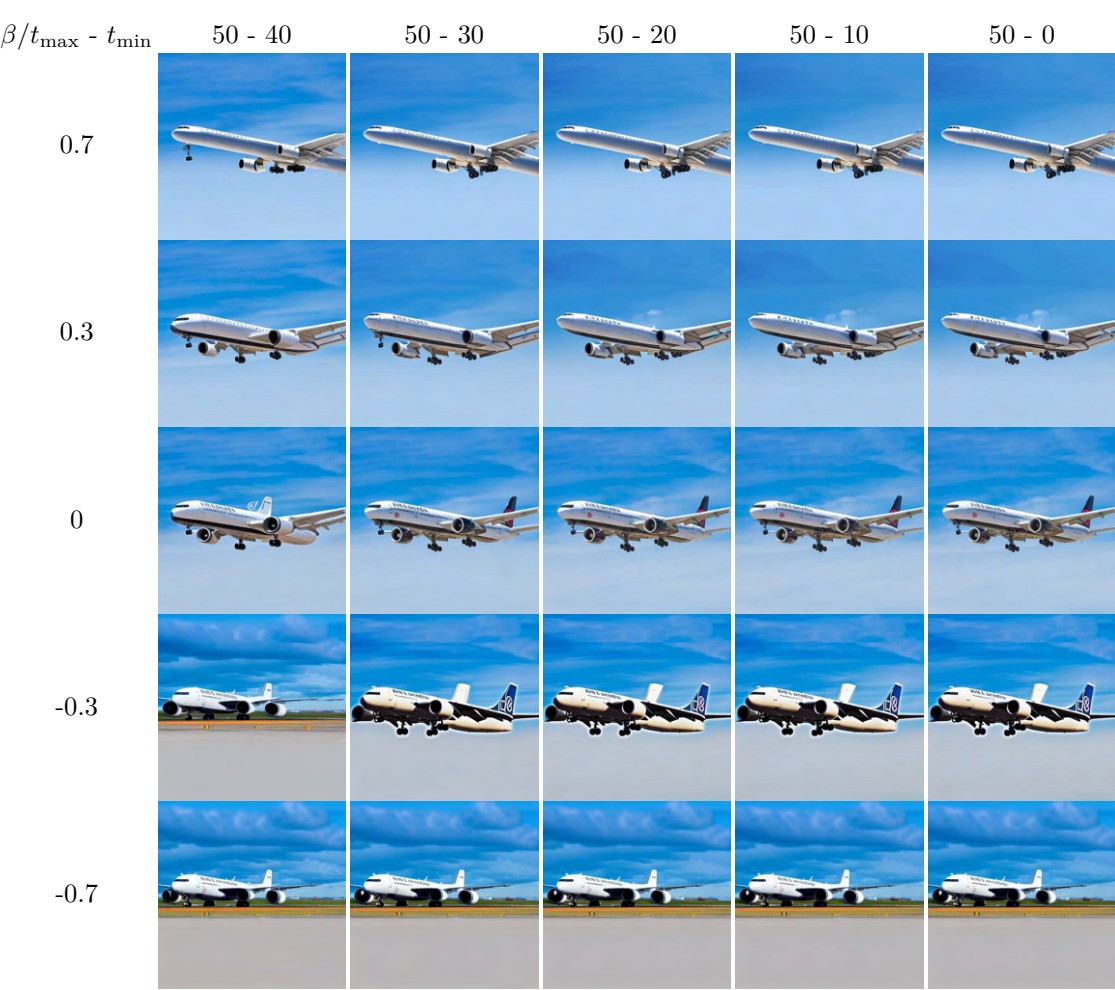

Figure 18: Results of MDP-$\beta$ using constant schedule when varying guidance scale $\beta$.

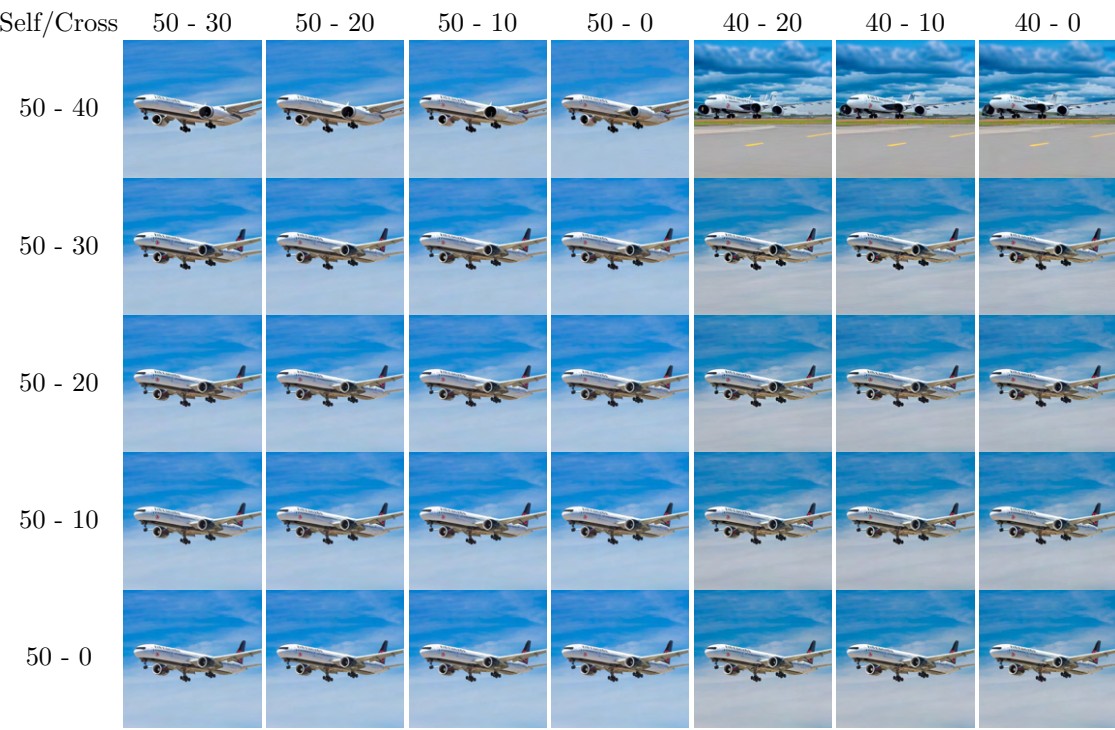

Figure 19: Results of Prompt-to-Prompt using constant schedule when varying the timestpes to inject self-attention maps and cross-attention maps. We show the results when we vary $t_{\max}$ and $t_{\min}$ for either injecting self-attention maps or cross attention maps.

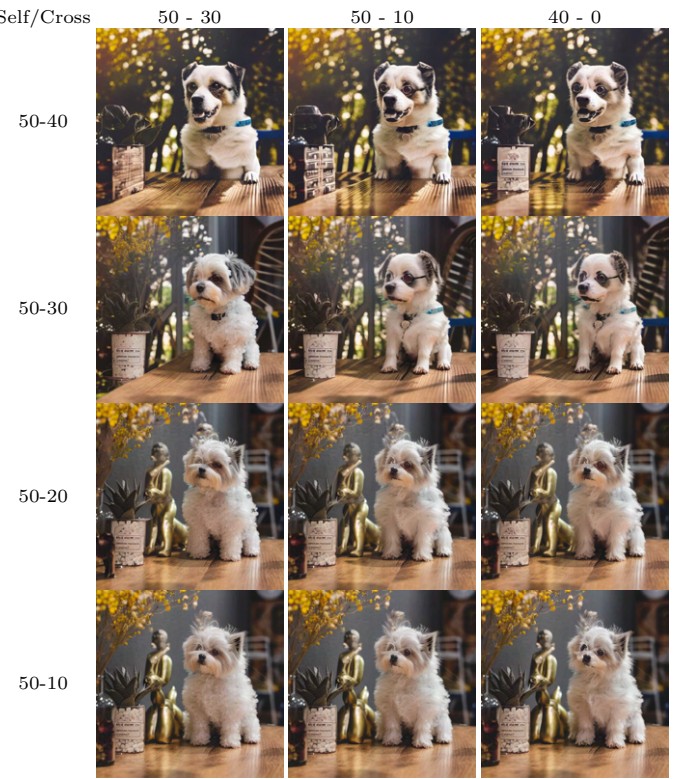

Figure 20: Additional results of Prompt-to-Prompt using constant schedule when varying the timestpes to inject self-attention maps and cross-attention maps. We show the results when we vary $t_{\max}$ and $t_{\min}$ for either injecting self-attention maps or cross attention maps.

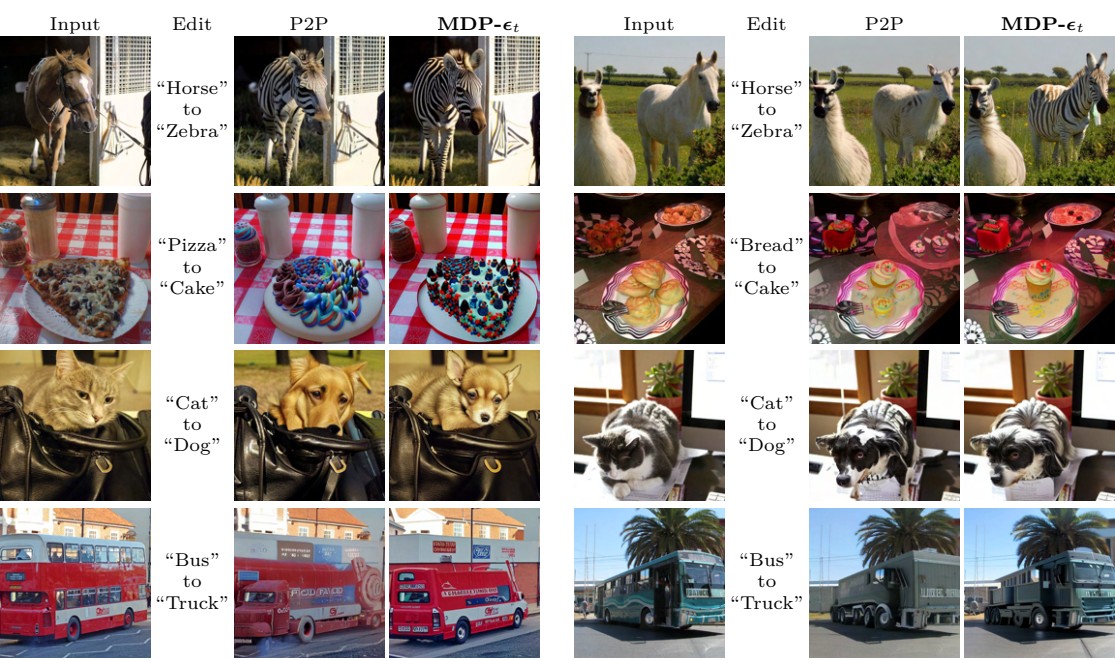

Figure 21: Results of changing object comparing Prompt-to-Prompt and **MDP-$\epsilon_t$**.

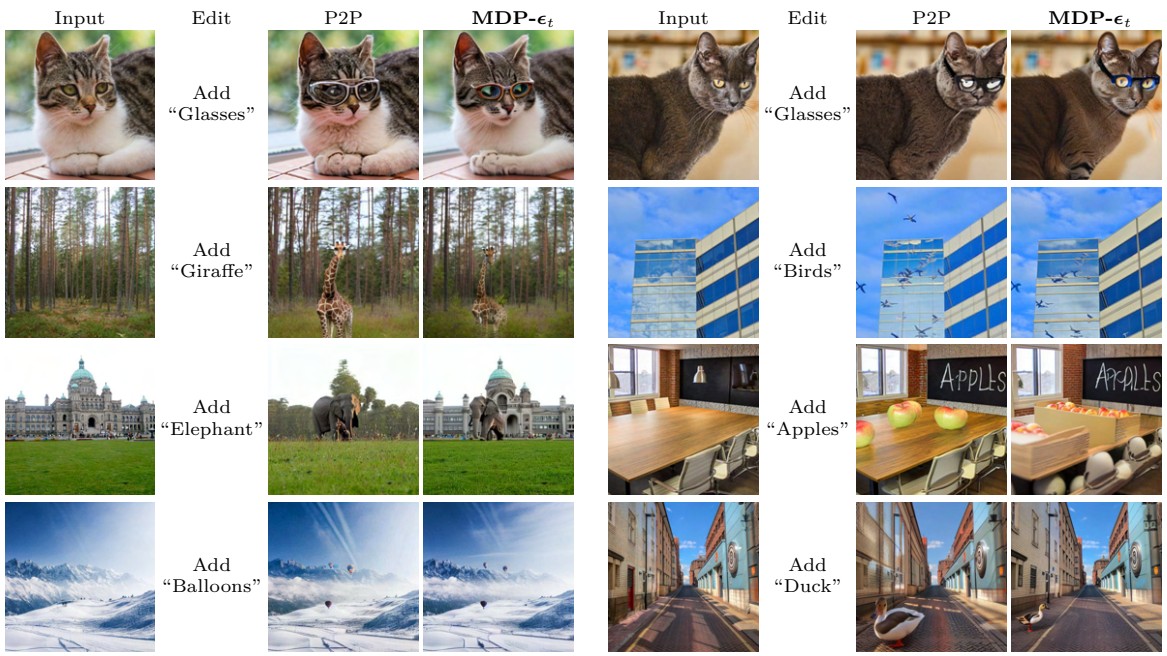

Figure 22: Results of adding object comparing Prompt-to-Prompt and **MDP-$\epsilon_t$**.

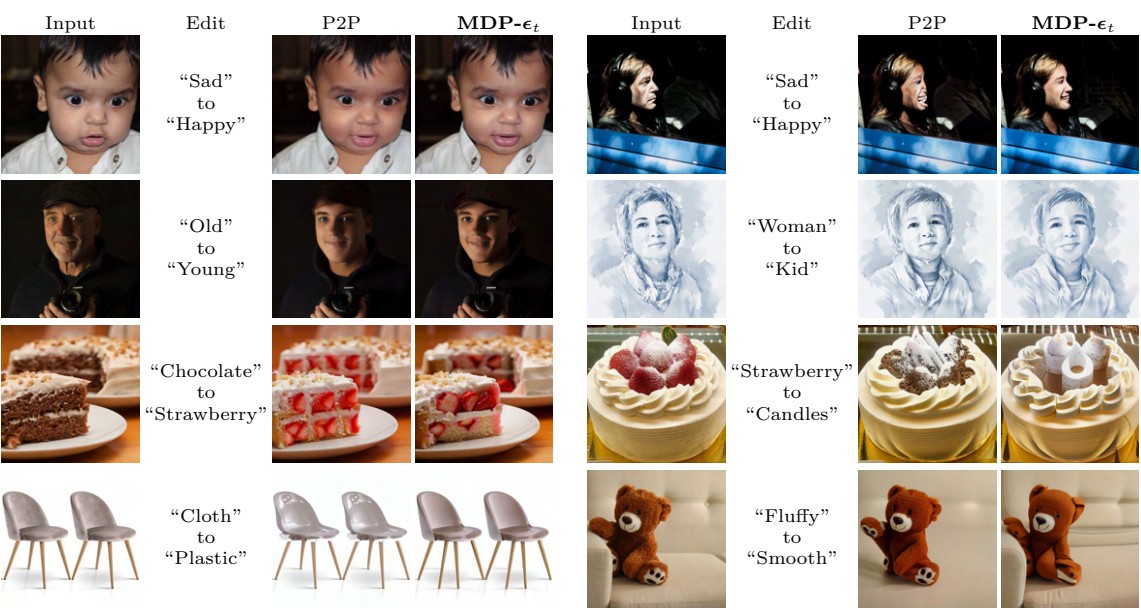

Figure 23: Results of changing attributes comparing Prompt-to-Prompt and **MDP-$\epsilon_t$**.

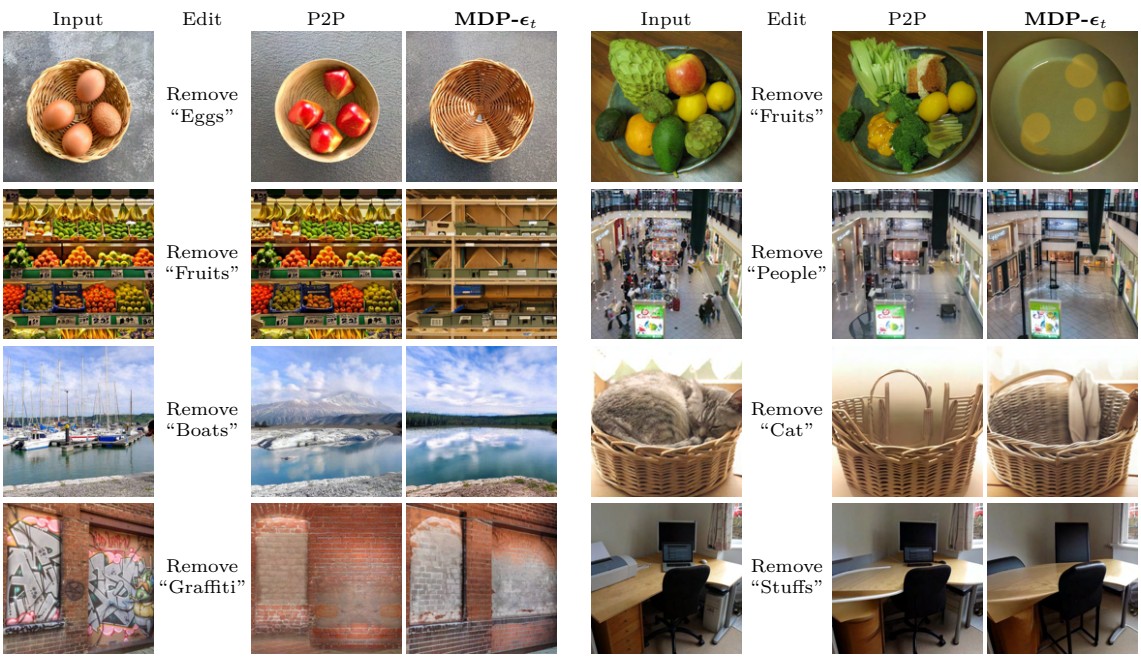

Figure 24: Results of removing object(s) comparing Prompt-to-Prompt and **MDP-$\epsilon_t$**.

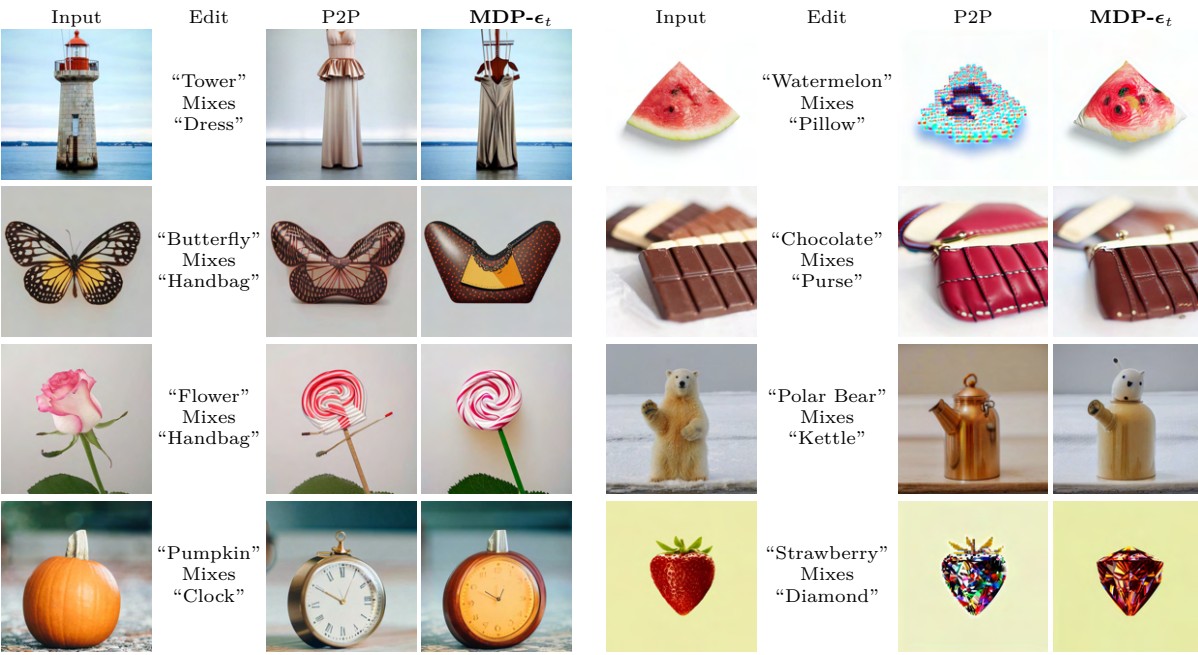

Figure 25: Results of mixing objects comparing Prompt-to-Prompt and **MDP-$\epsilon_t$**.

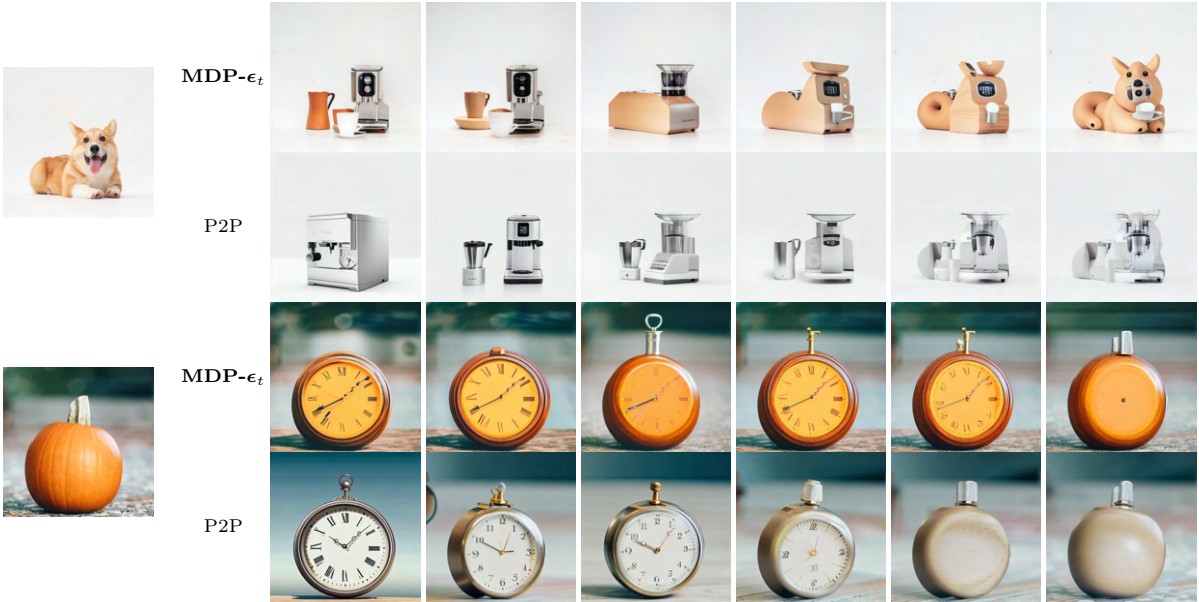

Figure 26: More intermediate results of mixing objects comparing **MDP-$\epsilon_t$** and P2P. In the first two rows we mix the input "corgi" with "coffee machine". In the second two rows we mix the input "pumpkin" with "clock".

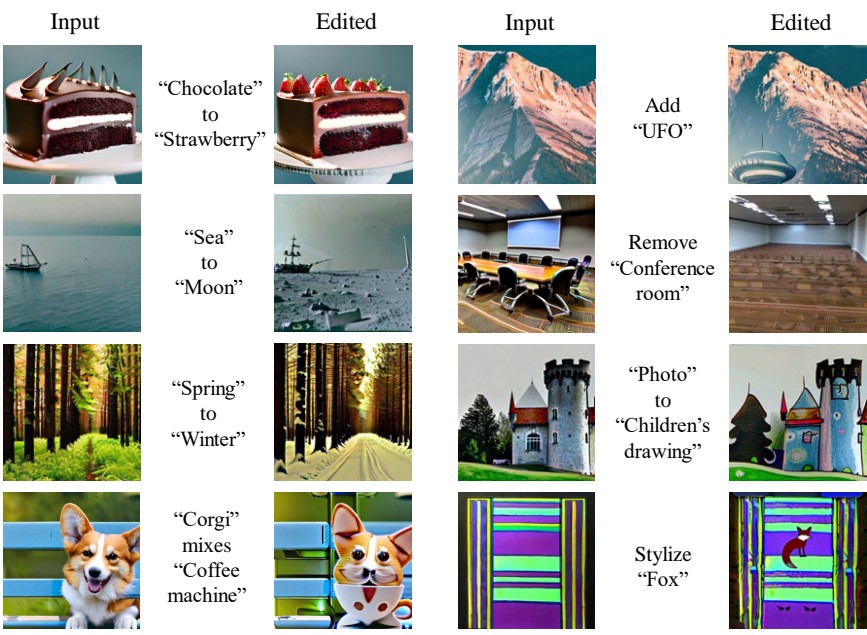

Figure 27: Editing results of **MDP-$\epsilon_t$** using Stable Diffusion 2.1. Note that all the input images are synthesized from Stable Diffusion 2.1.

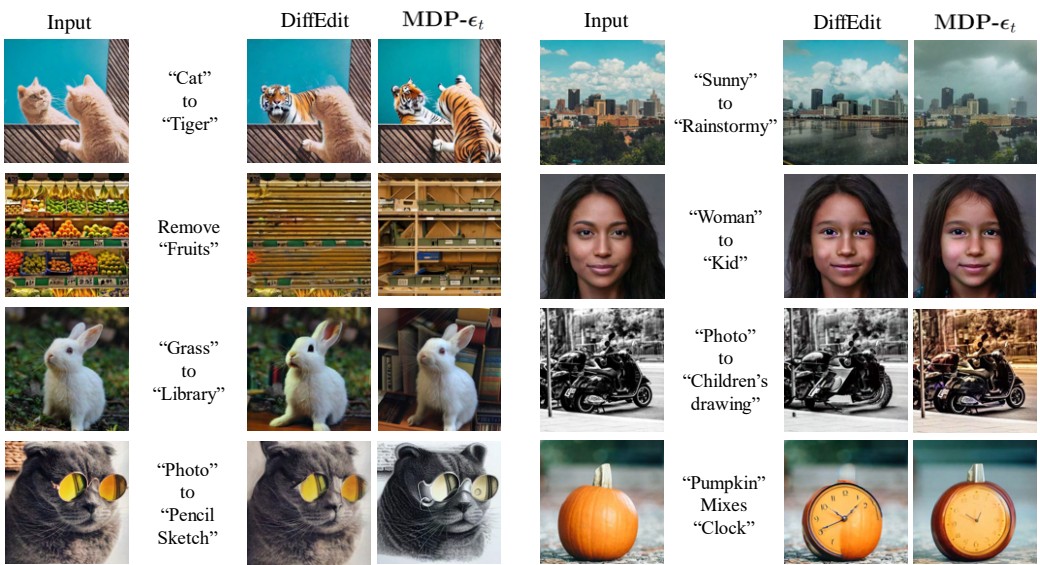

Figure 28: Comparison between the results of DiffEdit and **MDP-$\epsilon_t$**.

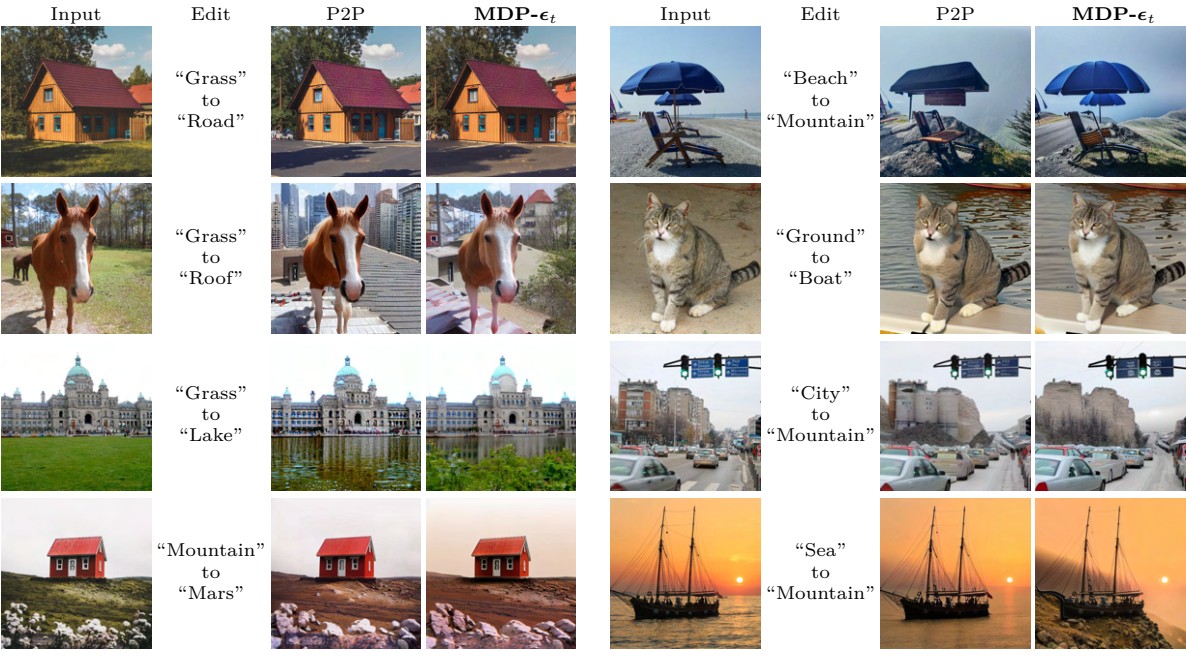

Figure 29: Results of changing background comparing Prompt-to-Prompt and **MDP-$\epsilon_t$**.

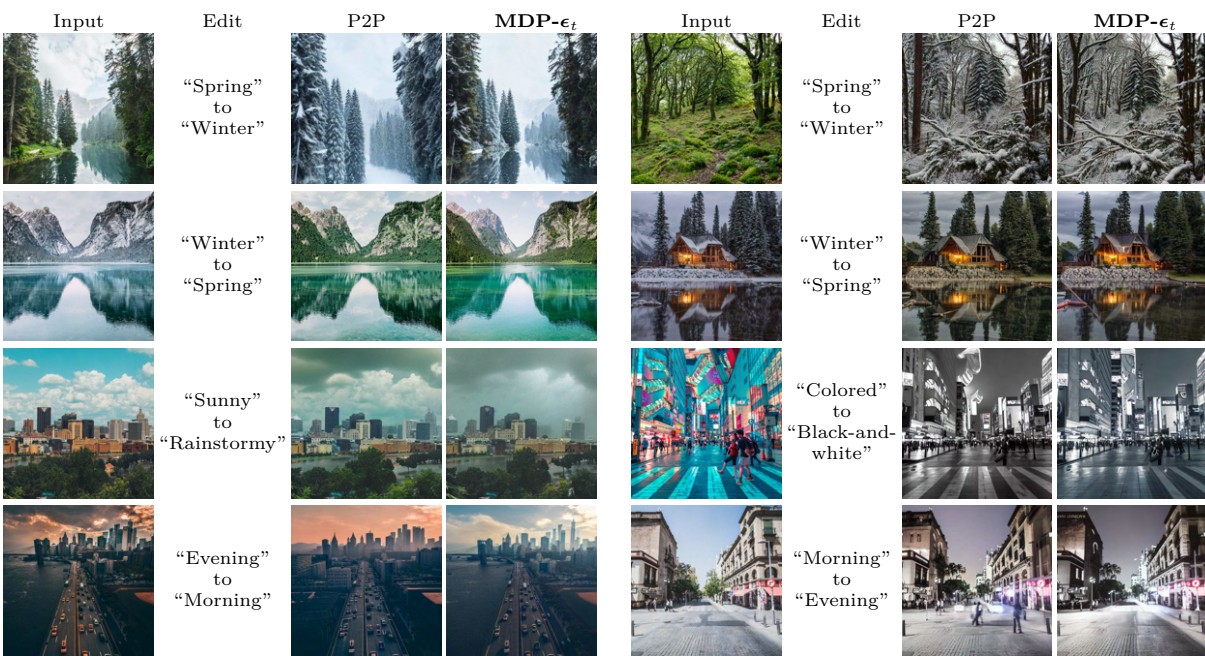

Figure 30: Results of in-domain transfer comparing Prompt-to-Prompt and **MDP-$\epsilon_t$**.

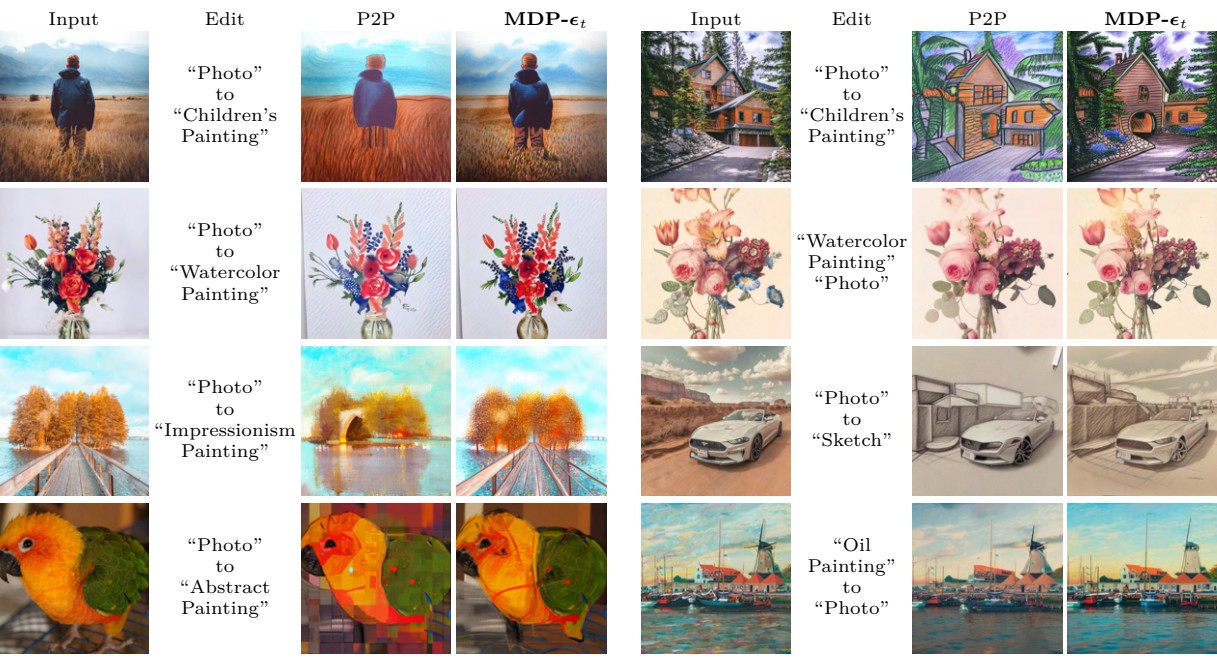

Figure 31: Results of out-domain transfer comparing Prompt-to-Prompt and **MDP-$\epsilon_t$**.

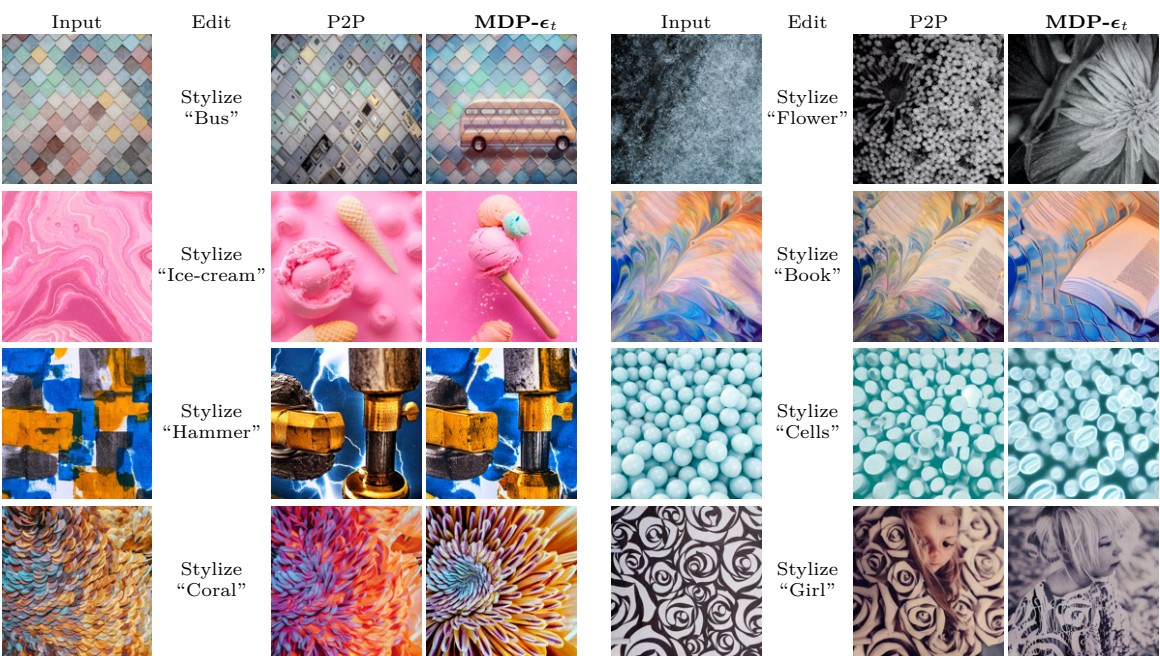

Figure 32: Results of stylization comparing Prompt-to-Prompt and **MDP-$\epsilon_t$**.

