# OpenReview forum: "MDP: A Generalized Framework for Text-Guided Image Editing by Manipulating the Diffusion Path"
_TMLR — Accepted by TMLR_

### Review · Reviewer_m17B · 2024-06-03

**Summary Of Contributions:**

This paper sets a general framework for the training-free text-guided image editing task. It shows that a sufficient amount of literature methods are special cases of that framework and it propose a novel method from that framework that achieves high quiality results.

**Audience:**

Yes

**Broader Impact Concerns:**

As I stated in W1 discussion about ethics is extremely important. The topics that need to be addressed are:

1) Biases and discrimination from the datasets of the pretrained models

2) Possible use for missinformation from the end user

3) Possible exploit of intellectual properties from the industry

**Claims And Evidence:**

Yes

**Requested Changes:**

Important for acceptance:

1) I think a discussion about the ethics of generative models should be included in the manuscript.

2) Also the availability of the initial condition should be discussed either with experiments or by arguing in the paper why this is not needed


minor typos, convensions, or notation choices:

1) Although the acronym MDP can be derived from the title, it is not defined in the text

2) In page 4 in the paragraph "Intermediate denoised output", second line: is x0(B) supposed to be c(B)?

3) In my experience "lerp" is quite unconventional as a math operation, would you consider to substitute lerp(A, B, w) with (1-w)A+wB?

4) References to tables or figures are not stating if they are for a table or for a figure. (page 3 second to last line "experiment in 2", page 6 before section 4 "method in 3", page 7 first line of 4.2 "results in 2")

**Strengths And Weaknesses:**

Strengths

S1) The paper is easy to follow. Most of the relevant information were quite clear even from the first read.

S2) The more general topic of generative models is quite important and relevant right now.

S3) The paper places the literature into a more general framework

S4) The paper shows that this framework can "produce" other, useful solutions to the task and the proposed method seems to work quite well

Weaknesses

W1) The paper is not discussing the ethical issues of generative models

W2) What happens if you don't have the condition c(A) available? Will this still work if you generate the c(A) from x0(A) through a captioner?

W3) minor typos, convensions, or notation choices

---

### Review · Reviewer_pdgE · 2024-06-05

**Summary Of Contributions:**

The authors introduce a generalized editing framework, MDP that encompasses five different types of manipulations: intermediate latent, conditional embedding, cross attention maps, guidance, and predicted noise. The MDP framework in general enough to encompass existing methods for image editing.

**Audience:**

Yes

**Claims And Evidence:**

Yes

**Requested Changes:**

See weaknesses

**Strengths And Weaknesses:**

**Strengths:**
Good quantitative and qualitative experimental results.

**Weaknesses:**
In general, the presentation of the paper is poor–-the citations appear strangely, multiple instances of cross-references are handled oddly, and there are formatting problems throughout. Further, there are grammatical errors and typos mixed throughout the paper.

The caption for Table 2 is uninformative, and there is not much discussion of these results. The CLIP directional similarity for DiffEdit is bolded, but it is actually the lowest of the methods. Figures 4 and 5 do not show any results from DiffEdit, while Table 2 includes these results.

Formatting problems:
- Citations always appear as: author (year)
- The text in Table 1 is too small
- References to figures/tables often just show as a number, while references to equations show as:  “Eq. equation [equation number]”
- There is a dangling period after the sentence following equation 3
- Equations 5 and 6 are split over two lines when there is space for them to fit on a single line
- Figure 4: “Chocolate” to “Strawberry” row of images is not aligned with the rest
- Figure 5: Stylize “Chair” row of images is not aligned with the rest
- Figures 12, 14, 16, etc. are too wide

---

### Review · Reviewer_QNkA · 2024-07-25

**Summary Of Contributions:**

This work discusses a unified diffusion-based image editing framework, consisting of a series of important existing diffusion-based image editing methods. It then proposes a new method for image editing, which interpolates noises conditioned on different prompts at inference time. This work compares the proposed method with existing editing methods both quantitatively and qualitatively, and discusses the impact of different hyperparameters, as well as the limitations.

**Audience:**

No

**Broader Impact Concerns:**

I do not have any concerns regarding the broad impact section.

**Claims And Evidence:**

No

**Requested Changes:**

Please address my concerns outlined in the previous weaknesses section, particularly the questions raised in the “Major Issues” part. Additionally, please carefully proofread the paper to enhance its readability.

**Strengths And Weaknesses:**

Strengths:
+ The paper attempts to unify some important diffusion-based image editing methods into a single framework.

Weaknesses:

Major issue:
1. The claimed contribution in this work appears largely exaggerated. Specifically, the contribution claimed mainly includes "a framework for a generalized design space ... that includes multiple existing methods as special cases” for diffusion-based image editing, consisting of five different methods: (1) Interpolating denoised outputs, (2) Interpolating condition embeddings, (3) Change cross attention maps, (4) Classifier-free guidance, and (5) Interpolating predicted noise. Among these parts, (1), (3), and (4) are clearly existing works, as also suggested in this paper. Actually, (2) is also a previous finding, for example in [1,2]. Only (5) is a newly introduced algorithm. In other words, only one (or arguably, two) of the five methods are not from existing works, and the other methods are re-implementations of previous works. If more than half of the design space consists of existing methods, they cannot be claimed as merely special cases of a newly proposed framework.
2. For the proposed method (5), it is important to discuss its differences with method (4) classifier-free guidance theoretically and carefully analyze the empirical differences in the paper because they are very similar. To start with, in method (4), the diffusion noise is a linear combination between  $\epsilon^{(A*)}$ and $\epsilon^{(B*)}$, while in method (5), the diffusion noise is an interpolation between $\epsilon^{(A)}$ and $\epsilon^{(*)}$. The only difference, if I understand correctly, is the noises used for interpolation. How does this difference lead to significant quantitative and qualitative differences in the experiment?
3. Evaluation results: The experiment does not demonstrate the advantages of method (5). Specifically, from Table 2, the quantitative results show that different methods have their own advantages, and the qualitative results also do not show significant benefits. More importantly, in Sec. 4.2, the paper claims an advantage in user studies and states that the details of the user study will be in the Supplementary materials, but I couldn’t find them there. The description of the user study in the main text is ambiguous and lacks many details, and it is not clear whether "our method" refers to method (5) or the whole framework.

Minor issue:
1. Some of the discussions in the related work section are misleading. For example, (a) In the “Image editing with diffusion models” section, paragraph 2, the blended diffusion and blended latent diffusion are organized as requiring a finetuning process. From my understanding, these two works do not require finetuning of diffusion models and should therefore be placed in the third paragraph.  (b) For prompt-to-prompt, the paper mentions its drawback as “prompt-to-prompt always requires a prompt together with an input image” and claims that “our edits do not require prompting for an input image.” However, in the methods section, the paper states that the proposed method will need a text-condition $c^{(A)}$ for null-text inversion. There is a contradiction: Does your method require a text prompt? I think the answer is yes, and the claims in the related work section are incorrect.
2. The experiments are done with stable diffusion 1.4, which is an outdated model. Why not use the latest stable diffusion model?
3. The proposed method requires proper selection of $t_{max}$ and $t_{min}$, which could be tedious.
4. Many potential typos. For example: (a) Figure 1 is not referred to in the main text; the text that mentions Figures 2 and 3 lacks “Fig.” (b) On page 6, when quoting equations, the text uses “Eq. Equation.” (c) In Table 2, the number 0.1851 seems wrongly highlighted; it is not the highest number. (d) On page 6, first paragraph, it says “we interpolate the predicted noise xt(A) …” while in the paper, $\epsilon$ is used to refer to noise, and $x_t$ refers to the latent image. (e) Figure 6 is inconsistent with the text: In Figure 6, the $\epsilon$ is copied from the $\epsilon^{(A)}$, while in the text, $\epsilon$ is interpolated between $\epsilon^{(A)}$ and $\epsilon^{(B)}$.

[1] Imagic: Text-Based Real Image Editing with Diffusion Models
[2] Uncovering the Disentanglement Capability in Text-to-Image Diffusion Models

---

> ### Author Response · Authors · 2024-07-28
> **Reply to reviewer QNkA**
>
> Major issues:
>
> 1. The main focus of this paper is not just the identification of new manipulations, but **the unification of existing and new approaches under a comprehensive framework**. We show that previous efforts made in different image editing approaches lie under the same design space. Unifying different methods under the same framework helps in understanding the landscape of diffusion image editing techniques.
>
>     Furthermore, the “predicted noise interpolation” manipulation which is a natural product after the analysis of the design space and has not been explored before, shows its merits in many image editing applications.
>
>     We claim existing methods as “special cases” because despite those methods manipulating the same components as proposed in our framework, we have proposed and analyzed a more generalized manipulation schedule in Appendix, which has not been explored by previous work. However, we will refine the wording to enhance the clarity of our contributions.
>
>
> 2. In our experiments, empirically we find that for MDP-$\beta$ (manipulation 4), the edited results are largely favored for the new condition c(B). That means we have to choose a very large $\omega$ for input condition c(A) to maintain the original layout. Even after putting a large weight on input condition c(A), we showed in Figure 4 that the edited images still ignore the layout from the input image compared to the images edited by other manipulations.
>
>     Theoretically, we suspect that it is because, in the early stage of image synthesis, the conditions greatly control how the layout of the edited image will look. For MDP-$\beta$, directly mixing the noises predicted by condition c(B) and c(A) will disturb the perseverance of the input layout, as after each mixing step, the whole generation direction iteratively shifts towards new condition c(B) more. However, for MDP-$\epsilon_t$, the mixed noises are the noise newly predicted by condition c(B) and the noise pre-generated using only condition c(A). This will make sure that the whole generation direction preserves more information from the input image A. **In a word, MDP-$\beta$ falls short of preserving input layout, while MDP-$\epsilon_t$ tends to keep more information from the input image and performs better.**
>
> 3. We agree that in several editing applications, such as replacing or adding objects (the first four rows in Figure 4) in the local editing, all manipulations other than MDP-$\beta$ perform well. However, when it comes to more difficult applications such as replacing background in the global editing (the first and second rows in Figure 5), other manipulations have difficulty while MDP-$\epsilon_t$ can still perform well. We acknowledge that MDP-$\epsilon_t$ does not overwhelmingly outperform the other manipulations. However, we emphasize that MDP-$\epsilon_t$ consistently demonstrates stable performance, and in some cases, it even surpasses the other manipulations across various editing applications discussed in the paper.
>
>     Quantitative metrics: We argue that although these metrics we choose can somehow reflect the quality of the edited images from certain aspects, they do not necessarily align with the human perspective. Also, the differences between different edited images can be subtle and may not be easily captured by the metrics. Thus we also provided a user study as a complementary evaluation.
>
>     User study: We apologize for not including details of the user study in the Supplementary Materials. We will update a new version with the corresponding details. In addition, “our method” refers to manipulation 5.
>
>
> Minor issues:
>
> 1. Paragraph 2 under subsection “Image editing with diffusion models”: The reason we refer to Blended Diffusion and Blended Latent Diffusion as methods that require finetuning is that there are optimization processes in both methods. However, we do agree that the optimization processes do not involve finetuning the whole diffusion model itself. We will change from “finetuning process” to “optimization process”.
>
>     Prompting of Prompt-to-Prompt and our method: We thank the reviewer for pointing out the misleading part. In fact, both Prompt-to-Prompt and our method can adopt Null-text inversion, which may not require a prompt that describes the input image, but an empty during the inversion process. We will clear up this confusion in the next version.
>
> 2. Outdated model: We will provide some results using more up-to-date diffusion models in the Supplementary Materials.
>
> 3. Proper selection of $t_{max}$ and $t_{min}$: We acknowledge that hyperparameters such as $t_{max}$ and $t_{min}$ have a great influence on the edited images. Empirically, we provided a set of recommended parameters for different manipulations in Table 3 and Table 4. However, we agree that in some difficult cases, extra tuning of the hyperparameters may be necessary to obtain an optimal result.
>
> 4. Typos: We will correct these typos in the next version.

---

> > ### Comment · Reviewer_QNkA · 2024-08-05
> > **Thanks for the revision**
> >
> > Thanks for the revision; it clarifies some of my previous concerns. After reading the revision and your comments, I still have some remaining questions:
> >
> > Major: Quantitative results (The previous 3rd major issue). I agree with you that the quantitative metrics, such as CLIP/LPIPS score, cannot reflect the real quality in many cases. This makes the human study results more important, as they possibly become the only effective quantitative measurement. (Because of this, it was very frustrating to not be able to see the human study details in the original submission.) I agree that the human study result shows advantages over the P2P method. My question is why the human results only compare P2P with your proposed method, but not other methods. I am interested in why you chose this method as the baseline in the human study and not other methods in your framework.
> >
> > Minor: I am sorry that I made a typo in the previous comment on the 4th minor issue, (e), where my confusion was in Figure 3, not Figure 6. I was confused that in Figure 3, the edited image (2nd row) directly copies noises from the first row. But in equation 9 and the corresponding text, the noise is interpolated between c(A) and c(B). I am wondering which one is correct - and possibly, there needs to be a fix for this inconsistency if I understand correctly.

---

> > > ### Author Response · Authors · 2024-08-06
> > > **Reply to Reviewer QNkA**
> > >
> > > Thank you for your further comments. Here we try to address your concerns:
> > >
> > > Major:
> > >
> > > At the time of this submission, we did not prepare a user study for the additional manipulations mentioned in our paper. We observed that MDP-$\epsilon_t$ demonstrates stable performance across various examples qualitatively. However, we agree that human evaluation may be the most effective quantitative method. Due to the limited author response time, we are unable to prepare a user study for the other manipulations at this moment. Nevertheless, if requested, we are happy to provide human evaluations for other manipulations in the next version of our work.
> > >
> > > Minor:
> > >
> > > Thank you for pointing out this issue. In Equation 9, we formulated the predicted noise in a generalized manner, where the noise is interpolated between $\epsilon(A)$ and $\epsilon(B)$ with a factor $\omega$. In our actual implementation, we fix $\omega=1$ for MDP-$\epsilon_t$, meaning we directly copy $\epsilon(A)$. In Figure 3, the noises from the first row are directly copied to the second row, which is consistent with our implementation. However, we acknowledge that Figure 3 should be presented in alignment with Equation 9. We will update Figure 3 accordingly in the next version.

---

### Author Response · Authors · 2024-07-29
**Paper revision**

We appreciate the valuable feedback from the reviewers. Here we revised the paper based on the comments:

- We added additional editing results using Stable Diffusion 2.1 in Figure 27 in the Supplementary Materials. (Reviewer **QNkA**)

- We added the details of user study in section H in the Supplementary Materials. (Reviewer **QNkA**)

- We fixed the related work section in the main paper. (Reviewer **QNkA**)

- We added a broader impact section in section A in the Supplementary Materials. (Reviewer **m17B**)

- We discussed the availability of the initial condition in section G in the Supplementary Materials. (Reviewer **m17B**)

- We added a visual comparison between DiffEdit and MDP-$\epsilon_t$ in Figure 28 in the Supplementary Materials. (Reviewer **pdgE**)

- We expanded the discussion of the quantitative results in section 4.2 in the main paper. (Reviewer **pdgE**)

- We fixed various typos and formatting issues in the main paper. (Reviewer **QNkA**, **m17B**, **pdgE**)

---

### Decision · Action_Editor_5kLr · 2024-09-02

**Recommendation:** Accept with minor revision

**Comment:**

This paper proposes a framework to unify several diffusion-based image editing methods. Based on this framework, this paper identifies a potentially overlooked image editing technique. The paper compares the new editing technique with existing methods and shows that it achieves better quality in many scenarios. The reviewers generally acknowledge the contribution of the unified framework, but they also raised concerns before the rebuttal period, such as over-claimed contributions (QNkA), ethical issues (m17B), and typos (all reviewers). During the rebuttal, the author made a good effort in addressing these concerns by revising contribution claims, adding discussions, and fixing the typos. Reviewers QNkA and m17B are satisfied with the modifications after the rebuttal. Additionally, reviewer pdgE provided important comments about the novelty and ethical implications. It is worth mentioning that novelty is not a necessary criterion for TMLR’s acceptance. Furthermore, the AE checked the discussion threads and found that the author provided ethics discussions in the thread for reviewer m17B. As such, the AE recommends the acceptance of this paper with recommended modifications: (1) Please properly add the ethics discussion and algorithm comparison discussion to the camera-ready paper; (2) Please fix the inconsistency between Figure 3 and Equation 9.

**Audience:**

Yes.

**Claims And Evidence:**

Yes.